

# Boolean logic algebra driven similarity measure for text based applications

Hassan I. Abdalla[1] and Ali A. Amer[2]

[1] College of Technological Innovation, Zayed University, Abu Dhabi, Abu Dhabi, United Arab Emirates
[2] Computer Science Department, Taiz University, Taiz, Yemen

## ABSTRACT

In Information Retrieval (IR), Data Mining (DM), and Machine Learning (ML), similarity measures have been widely used for text clustering and classification. The similarity measure is the cornerstone upon which the performance of most DM and ML algorithms is completely dependent. Thus, till now, the endeavor in literature for an effective and efficient similarity measure is still immature. Some recently-proposed similarity measures were effective, but have a complex design and suffer from inefficiencies. This work, therefore, develops an effective and efficient similarity measure of a simplistic design for text-based applications. The measure developed in this work is driven by Boolean logic algebra basics (BLAB-SM), which aims at effectively reaching the desired accuracy at the fastest run time as compared to the recently developed state-of-the-art measures. Using the term frequency–inverse document frequency (TF-IDF) schema, the K-nearest neighbor (KNN), and the K-means clustering algorithm, a comprehensive evaluation is presented. The evaluation has been experimentally performed for BLAB-SM against seven similarity measures on two most-popular datasets, Reuters-21 and Web-KB. The experimental results illustrate that BLAB-SM is not only more efficient but also significantly more effective than state-of-the-art similarity measures on both classification and clustering tasks.

Corresponding author
Ali A. Amer, aliaaa2004@yahoo.com

## INTRODUCTION

Over the last 10 years, Natural Language Processing (NLP) has been subjected to rapid development as new techniques and methods are continuously introduced to satisfy the ever-disseminating availability of data. This kind of development is clearly being mostly reflected in the fields of Information Retrieval (IR), Data Mining (DM), and Machine Learning (ML) in which several techniques used the similarity/distance measures for different purposes (*Zhang & Zuo, 2019*; *Gweon, Schonlau & Steiner, 2019a*; *Amer, 2020*; *Kogan, Teboulle & Nicholas, 2005*; *Amer, Abdalla & Nguyen, 2021*).

Several machine learning techniques have demonstrated a surpassing performance, in the NLP field, to handle the voluminous constantly-piling data and information on the internet. Among these techniques are clustering and classification which are still commonly used in almost all scientific fields, including text mining, information retrieval, web search, pattern recognition, and biomedical based text mining (*Amer & Abdalla, 2020*;

*Rachkovskij, 2017*; *Gweon, Schonlau & Steiner, 2019b*; *Kanungo et al., 2002*; *Holzinger et al., 2014*). For example, in *Holzinger et al. (2014)*, a detailed survey in biomedical-based text mining and classification was done, while stressing the importance of involving and improving similarity measures for classification tasks. Literature has long been stressing the performance of text clustering and classification which depends mainly on the similarity measures. Essentially, both tasks need highly-effective and maximally-efficient similarity measures to reach the desired rendering. However, finding suitable similarity measures for text clustering and classification is a challenging task. Efficiency and effectiveness are the basic characteristics each similarity measure should enjoy.

Generally speaking, in information retrieval, the documents are drawn as vectors in the vector space model (VSM) (*Amer & Abdalla, 2020*). In each document's vector, each cell refers to the value of the relative feature that corresponds to the term presence/absence. In this vector, this value is integer/real number represented by the weighting schema which is commonly falling into one of three cases: (1) term frequency (used in BoW) which indicates the appearance times of the relative term, (2) relative term frequency (TF) which is computed as the ratio between the frequency of term and the net number of appearances of all unique terms in the whole set of documents, (3) TF-IDF which is the most commonly-used schema in information retrieval (*Afzali & Kumar, 2017*). Occasionally, documents are broken into a large number of features represented in VSM, resulting in high sparse VSM. In other words, VSM would contain a rather low percentage of non-zero feature values. Consequently, the sparsity along with the dimensionality curse could have a severely negative impact on the performance of classifiers.

To tackle such challenges, in IR literature, a dozen of works have introduced several effective similarity measures for text clustering and classification (*Amer & Abdalla, 2020*; *Oghbaie & Mohammadi Zanjireh, 2018*; *Sohangir & Wang, 2017*; *Lin, Jiang & Lee, 2014*; *Shahmirzadi, Lugowski & Younge, 2019*; *Ke, 2017*; *White & Jose, 2004*; *Lakshmi & Baskar, 2021*; *Kotte, Rajavelu & Rajsingh, 2020*; *Thompson, Panchev & Oakes, 2015*). However, except for *Amer & Abdalla (2020)*, these studies proposed similarity measures without providing sufficient insights into run-time efficiency. In other words, these studies might introduce effective measures yet time-inefficient. Moreover, these measures, which are shown effective (*Amer & Abdalla, 2020*; *Oghbaie & Mohammadi Zanjireh, 2018*; *Sohangir & Wang, 2017*; *Lin, Jiang & Lee, 2014*; *Shahmirzadi, Lugowski & Younge, 2019*; *Lakshmi & Baskar, 2021*; *Robertson, 2004*) suffer from design complexity. Motivated by this, this work comes with the ultimate aim of finding an influential solution to the efficiency as well as the design complexity of those similarity measures. A Boolean logic algebra-driven similarity measure (BLAB-SM) is simply designed with the aim of its being significantly effective and highly efficient. BLAB-SM takes the presence and absence of each term as long as this term exists in either or both documents under consideration, making it highly competitive to give a robust classification. Seven similarity measures are thoroughly examined under diversified conditions. Using TF-IDF representation (*Zhao & Mao, 2018*; *Joulin et al., 2017*), the K-nearest neighbor (KNN), and the K-means algorithm, this work investigates each similarity measure with varying both the K values of KNN and the number of features of each dataset. Similar to *Amer & Abdalla (2020)* who

used the BoW model, we have used the TF-IDF model to assess all measures against low dimensional datasets by analyzing performance on (50, 100, 200, and 350 features) and high dimensional datasets (3,000, 6,000, and all features of each dataset). The performance of measures was studied profoundly to specify the measures that would yield the desired results in each K value of KNN and on each number of features. The key contributions, of this work, are as follows:

1. Presenting a new similarity measure whose behavior is driven from the Boolean logic algebra mechanism. It is named Boolean logic algebra-based similarity measure for text clustering and classification (BLAB-SM). Based on a rigorous experimental study, BLAB-SM has been shown a top performer by outperforming all the state-of-the-art measures concerning both effectiveness and efficiency. Indeed, BLAB-SM is one of the fastest measures comparing with all considered measures in this study including cosine, Euclidean, and Manhattan. The experimental results illustrate that BLAB-SM is not only more efficient but also significantly more effective than state-of-the-art similarity measures on both classification and clustering tasks.

2. Drawing a comprehensive and fair evaluation study for BLAB-SM against seven similarity measures. The performance of all similarity measures is benchmarked on web-KB and Reuters-21 datasets. Using TF-IDF, the KNN classifier, and the K-means clustering, a comparative study of the effectiveness and efficiency of these measures is made. Interestingly, this study has experimentally shown that the Jaccard similarity measure is an ineffective measure for TF-IDF-based document matching.

### Paper organization

The related works, including the methods compared, are covered in "Related WorK". "The Proposed Method" introduces the proposed similarity measure. Experimental settings are concisely presented in "Experimental Setup". The outcomes of experimental study are drawn in "Results". The most important points out of this study are articulated in "Discussion". Finally, "Conclusions and Future Work" concludes the paper and presents the avenues of future work.

## RELATED WORK

In IR literature, the vector space model (VSM) has widely been utilized to find the pairwise document similarity using the relative similarity measures. For example, as geometric measures, the Euclidean, Minkowski, Manhattan, and Chebyshev distances are utilized in (VSM) for text classification through finding the distance between each vector pair (*Heidarian & Dinneen, 2016*; *Cordeiro, Amorim & Mirkin, 2012*). In general, Manhattan distance is mostly more efficient than Euclidean on small datasets, yet it has long been recorded to have less accurate results in most of the text classification tasks particularly with THE sparse data. In our work, Euclidean was seen more efficient when run on web-kb and Reuters, though, That is due to the Manhattan is being contingent upon the rotation of the coordinate system, leading to its being disadvantageous for both document classification and clustering tasks (*Kumar, Chhabra & Kumar, 2014*). Meanwhile, Jaccard,

Ex-Jaccard, Kullback–Leibler divergences (KLD), and Bhattacharya coefficient were all used for the several tasks of ML and IR including text clustering and classification (*Amer & Abdalla, 2020*; *Tanimoto, 1957*; *Tata & Patel, 2007*; *Oghbaie & Mohammadi Zanjireh, 2018*, *François, Wertz & Verieysen, 2007*; *D'hondt et al., 2010*; *Li et al., 2017*; *Kullback & Leibler, 1951*).

On the other hand, Jaccard is always used to compute the ratio of the number of terms/points included in the document pair (or feature space) to the number of terms included in, at least, either one document/point. However, the Jaccard similarity measure does not utilize the term frequencies in the vector space model, making it an ineffective option for TF-IDF-based matching. Ex-Jaccard, therefore, comes to tackle the limitation of Jaccard when dealing with TF-IDF. However, Ex-Jaccard has long been shown time-inefficient. Both facts have been experimentally corroborated in our study of this paper. In contrast to most similarity measures/ distance metrics, KL divergence is asymmetric. That is, the KLD value from document d1 to document d2 does not equal the KLD value from d2 to d1. This fact contributes negatively to the performance of KLD when text classification considered. All of these distance/similarity measures are considered among the most efficient methods in IR and ML fields; yet, their performance has not reached the desired effectiveness, or even poor chiefly when the data are sparse or of high dimensions or both combined (*Subhashini & Kumar, 2010*; *Li & Han, 2013*). On the other hand, Cosine similarity, like Euclidean distance, is one of the most widely-applied similarity measures for text clustering and classification (*Amer & Abdalla, 2020*; *Arivarasan & Karthikeyan, 2019*; *Zhao & Karypis, 2002*). It seeks to find the cosine angle between vectors of each document pair. Cosine has long been shown to be effective and efficient at the same time. However, cosine suffers two limitations: (1) its effectiveness and efficiency have been degraded drastically when applied on high-dimensional datasets, or run on datasets of overlapping classes (*Amer & Abdalla, 2020*), (2) cosine is more suitable for text mining when data are a symmetric. Otherwise, Cosine might be either biased or less accurate (*Afzali & Kumar, 2017*).

To tackle this limitation, therefore, IR and ML literature has still been introducing new similarity measures. Off the most recently-published similarity measures are the set theory-based similarity measure (STB-SM). STB-SM was proposed in *Amer & Abdalla (2020)* for text classification and clustering, and compared against seven similarity measures in the context of the bag of word model. STB-SM was proven maximally effective and significantly efficient. In *Heidarian & Dinneen (2016)*, a geometric measure was introduced to find the similarity degree between each document pair. Depending on the information theory (*D'hondt et al., 2010*; *Aslam & Frost, 2003*), an Information-Theoretic measure for document Similarity (IT-Sim) was proposed. In the same line, in *Sohangir & Wang (2017)*, a similarity measure, named Improved Sqrt-Cosine similarity (ISC), was developed for text classification, and proven effective. In the same context, the pairwise document similarity measure (PDSM) was proposed in *Oghbaie & Mohammadi Zanjireh (2018)*. PDSM took into account the feature weights and the number of features found in, at least, one document. Another popular similarity measure for text processing, SMTP, was

presented in *Lin, Jiang & Lee (2014)* and shown highly effective when run against the state-of-art similarity measures including cosine, Euclidean and Sim-IT.

Even though the literature has presented a good number of similarity measures, the problem is that these proposed measures were either seen effective like PDSM, SMTP, and ISC; yet, they were time-inefficient. On the other extreme, other measures were seen efficient like Euclidean and Manhattan, yet are not as effective as PDSM, ISC or SMTP. As a compromised solutions, Cosine and STB-SM were described with their being closely effective and reasonably time-efficient. SYB-SM was seen much more effective (*Amer & Abdalla, 2020*), though. Given the drawn-above limitations of previous similarity measures, our work of this paper endeavors to introduce a novel similarity measure that would tackle these limitations while shrinking the trade-off between efficiency and effectiveness to the greatest extent. The proposed measure behaves based on Boolean logic algebra basis, called BLAB-SM, has been developed, and experimentally shown to outperform all compared state-of-art effectively and efficiently.

## The compared methods

In the following, all compared similarity measures and distance metrics are described. Having two documents document1 and document2 whose TF-IDF of their "n" terms (w.t) have been saved in both vectors: $doc_1(w.t_{11}, w.t_{12,...}, w.t_{1n})$ and $doc_2(w.t_{21}, w.t_{22, ...}, w.t_{2n})$ respectively, the considered seven similarity measures or distance metrics [29–32] are listed as follows:

## Euclidean distance (ED)

ED computes the distance between each point pair in N-dimensional space. It is define by the following equation:

$$D_{Euc}(doc1, doc2) = \sum \sqrt{(doc_{11-}doc_{12})^2 + (doc_{21-}doc_{22})^2 + \ldots (doc_{n1-}doc_{n2})^2} \qquad (1)$$

## Cosine measure

The pairwise similarity is found between each document pair using both the dot product as well as the magnitude of both vectors $doc_1$ and $doc_2$.

$$Sim_{Cos}(doc1, doc2) = \frac{\sum_{i=1}^{n} doc_{i1} * doc_{i2}}{\sqrt{\sum_{i=1}^{n} doc_{i1}^2} * \sqrt{\sum_{i=1}^{n} doc_{i2}^2}} \qquad (2)$$

## Jaccard similarity measure

It is one of the most widely used similarity measures. However, the problem with Jaccard as we found experimentally is that Jaccard is highly reliant on the common values which are, unlike BoW representation in which Jaccard is working well, barely exist in

TF-IDF representation. This reliance factor makes Jaccard a very poor option when used for TF-IDF-based document matching. This measure is defined by next equation.

$$Sim_{jaccard}(doc1, doc2) = \frac{doc1 \cap doc2}{doc1 \cup doc2} \tag{3}$$

### Extended Jaccard

The extended Jaccard (ex-jaccard, in short) can be considered as the inverse of Jaccard coefficient, and its equation is drawn as follows:

$$Sim_{Ex-jaccard}(doc1, doc2) = \frac{doc1.doc2}{|doc1|^2 + |doc2|^2 - (doc1.doc2)} \tag{4}$$

### Manhattan

This distance metric computes the sum of absolute differences between the coordinates of vectors of document pair. It is defined as follows:

$$Manhattan - distance(doc1, doc2) = \sum_{i=1}^{n} |doc1_{i1} - doc2_{i2}| \tag{5}$$

### Kullback leibler divergence (KLD)

KLD finds the difference between the probability distributions, and is defined as follow:

$$Sim_{KL}(doc1, doc2) = \sum_{i=1}^{n} Pdoc1(doc_{i1}) * \log\left(\frac{Pdoc1(doc_{i1})}{Pdoc2(doc_{i2})}\right) \tag{6}$$

where $Pdoc_1(doc_{i1})$ is the value of $i^{th}$ feature of $doc_1$ and $Pdoc_2(doc_{i2})$ is the value of $i^{th}$ feature of $doc_2$.

### SMTP

This similarity measure was presented in *Lin, Jiang & Lee (2014)* for text clustering and classification. It considers two key scenarios when the feature exists in both documents $doc_1$ and $doc_2$, and when the feature is absent in the pair. It was defined as follows:

$$Sim(doc1, doc2) = \frac{F(doc1, doc2) + \lambda}{1 + \lambda} \tag{7}$$

where

$$F(doc1, doc2) = \frac{N_{star}(doc1, doc2)}{N_{union}(doc1, doc2)}$$

$$N_{star}(doc1, doc2) = \begin{cases} 0.5\left(1 + exp^{-\left(\frac{doc_{i1} - doc_{i2}}{var}\right)^2}\right) & if \ doci1.doci2 > 0 \\ 0 & if \ doc_{i1} = 0, \ doc_{i2} = 0 \\ \lambda & Otherwise \end{cases}$$

And

$$N_{union}(doc1, doc2) = \begin{cases} 0, & if\ doci1 = 0, doci2 = 0 \\ 1, & Otherwise \end{cases}$$

where λ is a constant that was fixed at values of (1) and (0.01), while conducting our experiments, for classification and clustering respectively. The variable var indicates the standard distribution of all non-zero features. The values of (1) and (0.01) were not randomly chosen for λ. They were selected based on the setting asserted and used in *Lin, Jiang & Lee (2014)* as the optimal option to compare SMTP with other measures for both classification and clustering.

## THE PROPOSED METHOD

### Motivation

As mentioned earlier in the drawn-above sections, the text classification and clustering is tricky task chiefly when big data is set to be handled. Furthermore, even though the literature is full of effective similarity measures that have a good performance, the necessity for efficient and effective similarity measures is still incomplete. While there have been some efficient measures like Euclidean and Manhattan, these measures have been shown of poor performance when applied on texts of middle, big, or voluminous-sized datasets. On the other hand, there have been efficient and effective measures like cosine; however, cosine measure does not reach the desired performance. Therefore, several works in literature have been presenting new effective measures like PDSM, ISC, SMTP, and STB-SM. While PDSM was seen as a rather time-inefficient measure (*Amer & Abdalla, 2020*), SMTP has been seen time-inefficient as well in our work. On the other hand, these measures were seen much more effective than cosine; Yet not as efficient as cosine. The need to address the trade-off between efficiency and effectiveness motivates us mainly to enrich IR and ML with a new efficient and effective similarity measure (BLAB-SM). The newly-proposed measure has been proposed to shrink the already-mentioned trade-off. Experimentally, BLAB-SM is seen as effective as SMTP, and more efficient than cosine. It is also seen as efficient as Euclidean and Manhattan in most cases which is the ultimate objective of this work. Concisely, BLAB-SM comes to effectively fill the gap of efficiency problem from which some effective measures like SMTP suffers, while maintaining, if not outperforming in some cases, the effectiveness reached by SMTP.

### BLAB-SM similarity measure

Three basic definitions for the Boolean logic algebra, which inspired us to propose the measure, are briefly highlighted before presenting our proposed measure, BLAB-SM. In doing so, BLAB-SM's explanation and analysis would be completely understood.

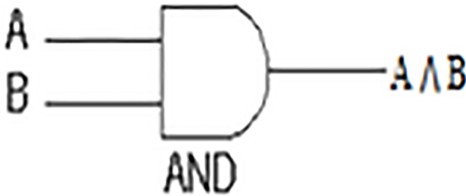

Figure 1 AND Gate.               

Table 1 A AND B table truth.

| A | B | A ∧ B |
|---|---|---|
| 0 | 0 | 0 |
| 0 | 1 | 0 |
| 1 | 0 | 0 |
| 1 | 1 | 1 |

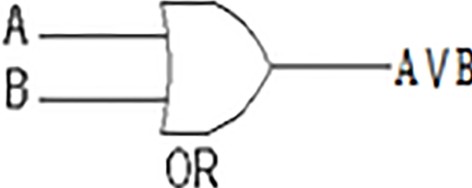

Figure 2 OR Gate.               

Table 2 A OR B table truth.

| A | B | A ∨ B |
|---|---|---|
| 0 | 0 | 0 |
| 0 | 1 | 1 |
| 1 | 0 | 1 |
| 1 | 1 | 1 |

### The logical gates

Strictly speaking, the digital systems are constructively defined by using the logic gates. In general, the basic gates are the AND, OR, NOT gates. The basic operations are described below along with their truth tables as follows:

The AND gate is an electronic circuit that yields an output (1) only and only if all its inputs (A and B, in Fig. 1, Table 1) are (1). This symbol (∧) is used to show the AND operation, i.e. A ∧ B. AND gate in our measure comprises only and only the shared features.

Like AND gate, the OR gate is also an electronic circuit. However, the OR gate yields an output (1) if and only if one or more of its inputs (A and B, in Fig. 2, Table 2) are (1).

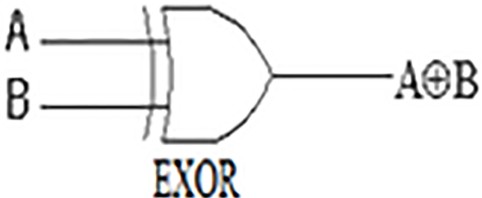

**Figure 3** EXOR Gate.               

**Table 3 A EXOR B table truth.**

| A | B | A ⊕ B |
|---|---|-------|
| 0 | 0 | 0 |
| 0 | 1 | 1 |
| 1 | 0 | 1 |
| 1 | 1 | 0 |

This symbol (∨) is used to show the OR operation, i.e., A ∨ B. OR gate in our measure concerns both the shared and non-shared features at the same time.

The 'Exclusive-OR' gate is a circuit that yields an output (1) if either one, but not both, of its two inputs are (1). The symbol with an encircled plus sign (⊕) is utilized to signal the Ex-OR operation, i.e., A ⊕ B (see A and B in Fig. 3 and Table 3). EXOR gate in our measure concerns only and only the non-shared features so as to non-shared features are given a chance to contribute in calculating similarity degree.

### The Boolean logic algebra based similarity measure (BLAB-SM)

Based on the already drawn concepts, the Boolean algebra based similarity measure (BLAB-SM) is defined as follows:

$$BLAB - SM(doc1, doc2) = \alpha \times X + \beta \times Y \tag{8}$$

Both α and β are parameters that have experimentally been tuned, as drawn in the results section, just to find their best values to detect the highest possible similarity. Although X is concerned with finding similarities between all non-shared features, it finds similarity between all shared features implicitly as well. It finds similarities based on logical sensing of the differences between both documents under consideration. All documents are logically treated and processed the same way logical gates work. On the other hand, Y emphasizes similarity through finding likeness between documents across all shared features only. Assuming having two documents which are document1 and document2 which are represented by vectors $doc_1$ and $doc_2$ in the vector space model using TF-IDF representation. Using $doc_1$ and $doc_2$ with "n" terms, X and Y are defined as follows:

$$X = \left( 1 - \frac{\sum_{I=1}^{n} (doc_{i1} \oplus doc_{i2})}{\sum_{I=1}^{n} (doc_{i1} \vee doc_{i2})} \right)$$

$$Y = \left( \frac{2 \times \left( \sum\limits_{I=1}^{n} (doc_{i1} \wedge doc_{i2}) \right)}{\left( \sum\limits_{I=1}^{n} doc_{i1} \right) + \left( \sum\limits_{I=1}^{n} doc_{i2} \right)} \right)$$

Meanwhile, $doc_1$ or $doc_2 = 1$, if $doc_1$ or $doc_2 \geq 1$; Otherwise, 0.

## BLAB-SM analysis

In this sub-section, we concisely, simply and informatively analyze the cases of the proposed measure. Assuming that X consists of $X_1$ and $X_2$, and Y consists of $Y_1$, $Y_2$, and $Y_3$. Accordingly, the X and Y equations are re-drawn as follows:

$$X = \left( 1 - \frac{X_1}{X_2} \right)$$

where:

$$X_1 = \sum_{I=1}^{n} (doc_{i1} \oplus doc_{i2})$$

$$X_2 = \sum_{I=1}^{n} (doc_{i1} \vee doc_{i2})$$

Similarly:

$$Y = \left( \frac{2 \times Y1}{Y2 + Y3} \right)$$

where:

$$Y_1 = \sum_{I=1}^{n} (doc_{i1} \wedge doc_{i2}),$$

$$Y_2 = \sum_{I=1}^{n} doc_{i1}, \, and \, Y_3 = \sum_{I=1}^{n} doc_{i2}$$

The perfect dissimilarity case:
This case happens when:

$X_1 = X_2 \Rightarrow$ X = zero, because X would equal $(1 - 1 = 0)$

$\because X_1 = X_2 \Rightarrow \quad Y_1 = zero \, (it \, is \, here \, AND \, gate), \quad X_1 = 1 \, since \, \frac{X_1}{X_2} = 1$

$\Rightarrow Y = \frac{2 \, x \, zero}{Y_2 + Y_3} = zero, \, regardless \, of \, Y2 \, and \, Y3 \, values$

$\Rightarrow \quad BLAB - SM \, (doc1, \, doc2) = \alpha \times X + \beta \times Y = \alpha \times zero + \beta \times zero = zero$

Example (perfect dissimilarity case): assuming we have doc1 (3, 0, 1) and doc2 (0, 2, 0). By the applying the perfect dissimilarity scenario, we find that BLAB-SM=zero, for both documents (1, 0, 1) and (0, 1, 0), which is logically true since there is no shared feature exist.

The average similarity:

$$X_2 > X_1 > zero \Rightarrow Y_1 > \text{zero} \Rightarrow 1 > X > zero, \text{ and } 1 > Y > zero \Rightarrow 1 > \text{BLAB} - \text{SM (doc1, doc2)} > zero$$

where (1) is the upper bound and (0) is the lower bound.

Example (Average similarity): assuming we have doc1 (3, 1, 1) and doc2 (6, 2, 0). By applying the average case scenario on both documents (1, 1, 1) and (1, 1, 0), BLAB-SM would have a value of roughly (0.73) which is bigger than zero and less than 1. It is worth indicating that 0.73 is logically reasonable than the cosine value which would reach 0.95. In fact, this is one novelty of our measure as similarity has never been exaggerated like what is done with the most state-of-art measure. Our measure allows non-zero non-shared features to have an implicit contribution to the similarity computation. Therefore, BLAB-SM takes the presence and absence of all features into consideration effectively.

The perfect similarity case:

$$X_1 = zero \Rightarrow X = 1 - \frac{\text{zero}}{X_2},$$

$X = 1$, regardless of the value of $X_2$
$\because X_1 = zero \Rightarrow Y_1 = \overline{X_1}$
$\Rightarrow Y_1 = 1, since Y_1 = complement (X_1) \Rightarrow Y = \frac{2 \text{ x } 1}{Y_2 + Y_3}$
$\because Y_1 = 1 \Rightarrow Length (doc1) = Length (doc2) \Rightarrow Y_2 = Y_3$
$\Rightarrow Y = \frac{2}{2xY_2 (or 2xY_3)} = \frac{2}{2} = 1$

Example (perfect similarity Case): assuming we have doc1 (3, 3, 3) and doc2 (3, 3, 3), or doc1 (1, 1, 1) and doc2 (1, 1, 1). By applying the best case scenario, we find that BLAB-SM=1 which is logically true as both documents are equal and equivalent.

## Similarity measure characteristics

Six characteristics should be defined on each similarity measure so this measure could be classified as good measure (*Haroutunian, 2011*; *Amigó et al., 2020*). These characteristics are given as follows:

**Characteristic 1:** The presence or non-presence of the targeted feature is more important than the difference between the values connected with the present feature.

Assuming we have $doc_1$ (3, 1, 1) and $doc_2$ (6, 2, 0). Then, the binary vectors, (1, 1, 1) and (1, 1, 0), of both documents are more important than the integer values linked to the values of features themselves in both documents. Let us take feature3 ($f_3$), we can say that feature3 has no link with $doc_2$ while it has a link with $doc_1$. In such case, both documents are dissimilar with respect to feature 3. So, we can conclude that feature3 has more weight in deciding similarity between $doc_1$ and $doc_2$ than $f_1$ and $f_2$ which are exist in both

documents. In fact, BLAM-SM focus in this issue when similarity is being calculated as behavior of BLAM-SM goes in the same way the logic of gates mechanism works.

**Characteristic 2:** The similarity degree should increase when the difference between the non-zero features values decrease. For example, having feature1 and feature2 as two features ($f_1$ and $f_2$) of $doc_1$ and $doc_2$, similarity ($doc_1$, $doc_2$) while $f_1$=10 and $f_2$=5 is higher than the similarity between $f_1 = 18$ and $f_2 = 4$.

**Characteristic 3:** The similarity degree should be increased when the number of present features grows. For example, having three vectors of three documents; $doc_1$ (1, 1, 0), $doc_2$ (1,1,1), $doc_3$ (1, 0, 0), the similarity ($doc_1$, $doc_2$) is far higher than similarity ($doc_1$, $doc_3$) due to the difference in the number of present or non-present features between all documents.

**Characteristic 4:** The document Pairs are highly different to each other if there have not had almost equivalent zero-valued features versus non-zero-valued features. For example, having two vectors of two documents $doc_1$ ($f_1$, $f_2$) = (1,0) and $doc_2$ ($f_3$, $f_4$) = (1,1), $doc_1.f_2$ and $doc_2.f_4$ are the main reason for maximizing the difference between both documents as $f_2 * f_4 = 0$, and, $f_2 + f_4 > 0$.

**Characteristic 5:** the symmetric properties should be met by each similarity measure. For example, the similarity degree between $doc_1$ (0, 1, 1, 0) and $doc_2$ (1,1,1,1) must be the same when $doc_2$(1,1,1,1) and $doc_1$ (0, 1, 1, 0) are addressed.

**Characteristic 6:** The value of distribution should contribute to the similarity degree between each document pair.

## EXPERIMENTAL SETUP

In the following, the experimental settings, algorithms, machine and data sets' descriptions, performance evaluation for both classification and clustering tasks are all elaborated.

For text clustering and classification applications, the feature extraction/selection and the pre-processing are crucial stages. First of all, text pre-processing was done to make the text process-able. Initially, the text was switched from upper case to lower case, and numbers, punctuations, stop words (common words) were removed. The extra spaces and symbols (like $, %) were also eliminated. For word stemming and text representation, python 3 was used to run the pre-processing using Nltk (Natural language toolkit) library. The ntlk word tokenizer was used for tokenization, and the Lemmatizing was done using the ntlk stem WordNetLemmatizer. Finally, stopword removal was done using the ntlk stopwords. Then, the vector space model (VSM) was utilized to represent features with TF-IDF representation (*Robertson, 2004*). The TF-IDF is the multiplication of term frequency (TF) by the inverse document frequency (IDF), and is defined as follows:

$$TF - IDF_{(t,d)} = tf(t, doc) * \log\left(\frac{N}{df + 1}\right) \tag{9}$$

$$where, tf(t, d) = \frac{count\ of\ term\ t\ in\ document\ doc}{number\ of\ words\ in\ document\ doc}$$

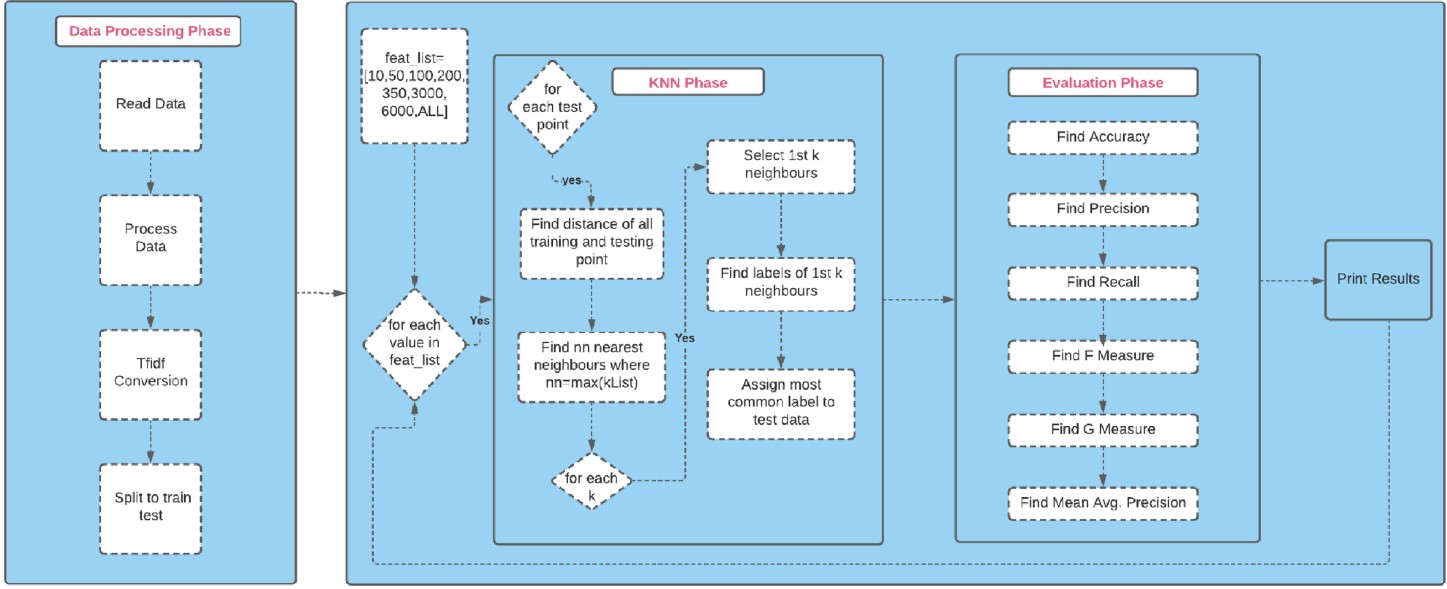

**Figure 4 The experimental design flow chart.**               

where df is the occurrence of t in all considered documents, and N is the total number of documents.

## KNN classifier

In general, KNN examines each test point based on its "*K*" closest "neighboring" points which are sorted and assumed to be the most similar points. Then, the test point is classified based on the voting technique. It takes the class label of the majority vote among all *K* neighboring training points in the feature space. The neighborhood is basically determined based on the used distance/similarity measure. Let us assume that $d_i$ is the training set, and $d_i^*$ is the testing set, c is the true class of training set, and c* is the predicted class for the testing set (c, c*= ..., m) where m is the number of classes. In the training classification process, only the true class c of each training set is used, and during testing, class c* is predicted for each testing set. On the other hand, when using the one-nearest neighbor rule, the predicted class of the testing set $d_i^*$ is set equal to the true class c of its nearest neighbor, where $w_i$ is the nearest neighbor to d* at the distance:

$$d\left(w_i, d_i^*\right) = \min_{j}\left\{d\left(w_j, d_i^*\right)\right\} \tag{10}$$

For KNN, the predicted test sample class $d_i^*$ is assigned to the most frequent true class among the K nearest training sets.

The experimental design flow chart, provided in Fig. 4, shows briefly how the KNN classifier was used to perform classification task successfully.

**Table 4 Splitting of documents among four classes in Web-KB dataset.**

| Class | Samples |
| --- | --- |
| Project | 504 |
| Course | 930 |
| Student | 1,641 |
| Faculty | 1,124 |
| Total | 4,199 |

**Table 5 Splitting of documents among eight classes in Reuters-R8 dataset.**

| Class | Samples |
| --- | --- |
| Cq | 2,292 |
| Crude | 374 |
| Earn | 3,923 |
| Grain | 51 |
| Interest | 271 |
| Money-fix | 293 |
| Ship | 144 |
| Trade | 326 |
| Total | 7,674 |

## Machine description

This work has been written in python language, and run on Processor Intel Core i5-3320M (2.6 GHz), RAM 4GB with OS Windows 7 (64 bit).

## Datasets description

Reuters-R8 (Table 4): it consists of 7674 documents with eight classes, and has 18308 features after it has been pre-processed.

Web-kb (Table 5): it consists of 4199 documents with four classes, and has 33,025 features after it has been processed. It is composed of web pages of computer science from universities: Cornell, Texas, Washington, and Wisconsin.

Both Tables 4 and 5 holds the description of both datasets used in this work. The data in each datasets was split into training and testing in ratio 2:1 (67%: 33%). It is worth indicating that these datasets are publicly available (https://gith ub.com/aliamer/Boolean-Logic-Algebra-Driven-Similarity-Measure-for-Text-Based-Applications/blob/main/Reuters%20%2B%20WebKB%20datasets.rar).

## Classification evaluation criterion
### *Accuracy*
It checks the sample total that are unmistakably classified out of the whole collection.

$$ACC = \frac{\text{True Positive} + \text{True Negative}}{\text{True Positive} + \text{True Negative} + \text{False Positive} + \text{False Negative}} \quad (11)$$

### Precision

It gives the whole number of items unmistakably identified as positive out of the whole items defined as positive.

$$PRE = \frac{\text{True Positive}}{\text{True Positive} + \text{False Positive}} \quad (12)$$

### Recall

It gives the whole number of items unmistakably identified as positive out of the actual positive

$$REC = \frac{\text{True Positive}}{\text{True Positive} + \text{False Negative}} \quad (13)$$

### F-method

It is a harmonic mean of precision and recall.

$$\text{F Score} = 2 * \frac{\text{Precision} * \text{Recall}}{\text{Precision} + \text{Recal}} \quad (14)$$

### G-method

It is used as a geometric mean of both precision and recall.

$$GM = \frac{\text{True Positive}}{\sqrt{(\text{True Positive} + \text{False Positive}) * (\text{True Positive} + \text{False Negative})}} \quad (15)$$

### Average mean precision (AMP)

It is the average of the averaged precision of all classes.

$$AMP = \sum_{n} (R_n - R_{n-1}) P_n \quad (16)$$

where $P_n$ and $R_n$ are the precision and recall at the $n^{th}$ threshold.

## Clustering evaluation criterion

### Purity

To check the purity of each cluster as it judges the coherence of a cluster.

$$Purity = \frac{1}{N} \sum_{i=1}^{k} max_j |c_i \cap t_j| \quad (17)$$

where N = number of data points, k = number of clusters, $c_i$ is a cluster in C, and $t_j$ is the classification which has the max count for cluster $c_i$.

### Entropy

It is used to measure the extent to which a cluster contain single class and not multiple classes.

$$EN = \sum_{i=1}^{c} ci * log(ci) \tag{18}$$

## RESULTS

Before endorsing the best values of parameters α and β, each parameter was thoroughly and carefully tested while its value varied from (0.1) to (0.9) with an increment of (0.1) each time. Varying values from 0.1 to 0.9 for both parameters was to test BLAB-SM on all different scenarios so the desired effect would not be missed. Based on the results of the thoroughly-performed experiments on several versions of BLAB-SM, we approved values (0.5; 0.5) for α and β respectively when running BLAB-SM for both classification and clustering tasks.

### Classification results

Fixing α and β on values (0.5; 0.5), this work examines all similarity measures thoroughly using KNN classifier and K-means clustering algorithm. The K value in KNN was changed in each run from (1) to (120) as given in the Appendix. Furthermore, the number of features was falling in one of these values (10, 50, 100, 200, 350, 3000, 6000, NF) where NF is the whole number of features. After that, the results are averaged for each measure on all K values as given in Tables 6–14. On the other hand, albeit it has long been proven effective when dealing with the BoW model, the Jaccard measure has been excluded due to its being proven ineffective option when dealing with TF-IDF representation (see the Appendix). That is because of the fact that Jaccard is heavily based on the common values between each considered pair. However, the common values with TF-IDF is far less than it is with BoW. This fact justifies the poor behavior of Jaccard when dealing with TF-IDF-based document matching. The bolded values in the next Tables signify the best values. For readability, Accuracy, Precision, Recall, F-Measure, G-Measure, and Average Mean Precision are represented by ACC, PRE, REC, FM, GM, and AMP respectively.

In Table 6, for Reuters dataset, the BLAB-SM similarity measure, Manhattan followed by KLD achieved the highest accuracy. Ex-jaccard, although it was not among the best performers but it still outperformed Euclidean, SMTP, and cosine respectively with regard to accuracy. On the other hand, KLD followed by BLAB-SM and Manhattan had the best FM and AMP with BLAB is better with FM. However, with regard to the Web-KB dataset, Manhattan, followed by KLD and Euclidean, met the highest accuracy. For FM and AMP criterions, Manhattan followed by Euclidean and cosine, outweighed all other measures. Thus, the best measures were BLAB-SM, Manhattan, and KLD on Reuters, and Manhattan followed by KLD and Euclidean on Web-KB.

**Table 6 Performance evaluation of all measures when NF = 10 – the averaged results (K = 1–120; +2).**

| Dataset | Reuters-8 | | | | | | Web-KB | | | | | |
|---|---|---|---|---|---|---|---|---|---|---|---|---|
| Similarity/criterion | ACC | PRE | REC | FM | GM | AMP | ACC | PRE | REC | FM | GM | AMP |
| Euclidean | 0.642 | 0.345 | 0.272 | 0.262 | 0.504 | 0.203 | 0.611 | 0.598 | 0.540 | 0.553 | 0.679 | 0.438 |
| Cosine | 0.638 | 0.282 | 0.266 | 0.255 | 0.499 | 0.199 | 0.614 | 0.592 | 0.542 | 0.553 | 0.682 | 0.442 |
| Ex-Jaccard | 0.680 | 0.278 | 0.254 | 0.244 | 0.487 | 0.197 | 0.484 | 0.454 | 0.401 | 0.377 | 0.569 | 0.329 |
| Manhattan | 0.710 | 0.344 | 0.288 | 0.286 | 0.521 | 0.218 | 0.618 | 0.620 | 0.548 | 0.562 | 0.685 | 0.448 |
| KLD | 0.696 | 0.361 | 0.294 | 0.277 | 0.525 | 0.219 | 0.612 | 0.629 | 0.523 | 0.527 | 0.669 | 0.437 |
| SMTP | 0.641 | 0.300 | 0.271 | 0.252 | 0.503 | 0.206 | 0.590 | 0.576 | 0.516 | 0.514 | 0.661 | 0.417 |
| BLAB-SM | 0.720 | 0.303 | 0.285 | 0.272 | 0.519 | 0.221 | 0.587 | 0.584 | 0.516 | 0.517 | 0.661 | 0.415 |

**Table 7 Performance evaluation of all measures when NF = 50 – the averaged results (K = 1–120; +2).**

| Dataset | Reuters-8 | | | | | | Web-KB | | | | | |
|---|---|---|---|---|---|---|---|---|---|---|---|---|
| Similarity/criterion | ACC | PRE | REC | FM | GM | AMP | ACC | PRE | REC | FM | GM | AMP |
| Euclidean | 0.817 | 0.635 | 0.572 | 0.586 | 0.745 | 0.465 | 0.684 | 0.747 | 0.605 | 0.634 | 0.728 | 0.522 |
| Cosine | 0.843 | 0.654 | 0.552 | 0.572 | 0.732 | 0.452 | 0.729 | 0.742 | 0.669 | 0.687 | 0.776 | 0.568 |
| Ex-Jaccard | 0.442 | 0.310 | 0.181 | 0.190 | 0.399 | 0.167 | 0.348 | 0.349 | 0.281 | 0.269 | 0.461 | 0.268 |
| Manhattan | 0.813 | 0.642 | 0.586 | 0.591 | 0.753 | 0.475 | 0.652 | 0.777 | 0.565 | 0.598 | 0.697 | 0.498 |
| KLD | 0.6245 | 0.602 | 0.282 | 0.320 | 0.505 | 0.242 | 0.236 | 0.527 | 0.316 | 0.217 | 0.496 | 0.292 |
| SMTP | 0.853 | 0.647 | 0.558 | 0.563 | 0.737 | 0.447 | 0.774 | 0.803 | 0.694 | 0.716 | 0.796 | 0.611 |
| BLAB-SM | 0.853 | 0.650 | 0.548 | 0.569 | 0.730 | 0.447 | 0.766 | 0.820 | 0.682 | 0.709 | 0.787 | 0.609 |

**Table 8 Performance evaluation of all measures when NF = 100 – the averaged results (K = 1–120; +2).**

| Dataset | Reuters | | | | | | Web-KB | | | | | |
|---|---|---|---|---|---|---|---|---|---|---|---|---|
| Similarity/ Criterion | ACC | PRE | REC | FM | GM | AMP | ACC | PRE | REC | FM | GM | AMP |
| Euclidean | 0.839 | 0.670 | 0.644 | 0.640 | 0.792 | 0.521 | 0.617 | 0.729 | 0.525 | 0.548 | 0.666 | 0.456 |
| Cosine | 0.870 | 0.686 | 0.592 | 0.610 | 0.760 | 0.493 | 0.747 | 0.776 | 0.685 | 0.705 | 0.788 | 0.593 |
| Ex-Jaccard | 0.852 | 0.632 | 0.523 | 0.544 | 0.713 | 0.433 | 0.464 | 0.555 | 0.372 | 0.365 | 0.544 | 0.325 |
| Manhattan | 0.847 | 0.686 | 0.638 | 0.624 | 0.788 | 0.517 | 0.536 | 0.772 | 0.426 | 0.429 | 0.587 | 0.386 |
| KLD | 0.555 | 0.659 | 0.211 | 0.232 | 0.432 | 0.197 | 0.394 | 0.386 | 0.260 | 0.166 | 0.442 | 0.256 |
| SMTP | 0.884 | 0.667 | 0.603 | 0.614 | 0.768 | 0.499 | 0.781 | 0.815 | 0.692 | 0.708 | 0.796 | 0.617 |
| BLAB-SM | 0.879 | 0.669 | 0.585 | 0.610 | 0.756 | 0.486 | 0.778 | 0.853 | 0.693 | 0.720 | 0.795 | 0.629 |

In Tables 7 and 8, it is obvious that, for Reuters dataset, BLAB-SM followed by SMTP and cosine, obtained the highest ACC. However, with FM and AMP, Manhattan, Euclidean, cosine and SMTP obtained the best values. On Web-KB dataset, SMTP, followed by BLAB-SM, and cosine, achieved the highest ACC, FM and AMP respectively. Thus, the top performer measures were SMTP, BLAB-SM, and cosine. From Table 9, on Reuters and Web-KB datasets, BLAB-SM, followed by SMTP and cosine, obtained the

**Table 9 Performance evaluation of all measures when NF = 200 – the averaged results (K = 1–120; +2).**

| Dataset | Reuters | | | | | | Web-KB | | | | | |
|---|---|---|---|---|---|---|---|---|---|---|---|---|
| Similarity/criterion | ACC | PRE | REC | FM | GM | AMP | ACC | PRE | REC | FM | GM | AMP |
| Euclidean | 0.821 | 0.738 | 0.607 | 0.637 | 0.766 | 0.533 | 0.578 | 0.742 | 0.470 | 0.482 | 0.624 | 0.414 |
| Cosine | 0.897 | 0.739 | 0.642 | 0.661 | 0.793 | 0.565 | 0.770 | 0.797 | 0.710 | 0.729 | 0.806 | 0.621 |
| Ex-Jaccard | 0.871 | 0.644 | 0.554 | 0.572 | 0.735 | 0.468 | 0.540 | 0.638 | 0.443 | 0.446 | 0.604 | 0.383 |
| Manhattan | 0.828 | 0.733 | 0.510 | 0.553 | 0.700 | 0.446 | 0.461 | 0.792 | 0.327 | 0.280 | 0.504 | 0.311 |
| KLD | 0.530 | 0.611 | 0.161 | 0.149 | 0.375 | 0.155 | 0.389 | 0.350 | 0.254 | 0.152 | 0.437 | 0.253 |
| SMTP | 0.897 | 0.721 | 0.635 | 0.648 | 0.789 | 0.545 | 0.800 | 0.839 | 0.725 | 0.748 | 0.818 | 0.651 |
| BLAB-SM | 0.900 | 0.719 | 0.629 | 0.650 | 0.785 | 0.545 | 0.807 | 0.874 | 0.727 | 0.757 | 0.820 | 0.669 |

**Table 10 Performance evaluation of all measures when NF = 350 – the averaged results (K = 1–120; +2).**

| Dataset | Reuters | | | | | | Web-KB | | | | | |
|---|---|---|---|---|---|---|---|---|---|---|---|---|
| Similarity/criterion | ACC | PRE | REC | FM | GM | AMP | ACC | PRE | REC | FM | GM | AMP |
| Euclidean | 0.780 | 0.785 | 0.551 | 0.608 | 0.727 | 0.501 | 0.567 | 0.737 | 0.451 | 0.452 | 0.610 | 0.394 |
| Cosine | 0.905 | 0.772 | 0.686 | 0.714 | 0.820 | 0.613 | 0.778 | 0.797 | 0.711 | 0.726 | 0.809 | 0.621 |
| Ex-Jaccard | 0.880 | 0.732 | 0.571 | 0.588 | 0.747 | 0.479 | 0.583 | 0.674 | 0.486 | 0.492 | 0.639 | 0.417 |
| Manhattan | 0.760 | 0.755 | 0.368 | 0.417 | 0.583 | 0.341 | 0.451 | 0.618 | 0.310 | 0.245 | 0.490 | 0.297 |
| KLD | 0.516 | 0.548 | 0.146 | 0.123 | 0.357 | 0.142 | 0.388 | 0.368 | 0.252 | 0.147 | 0.435 | 0.252 |
| SMTP | 0.902 | 0.778 | 0.657 | 0.679 | 0.803 | 0.570 | 0.812 | 0.852 | 0.741 | 0.765 | 0.830 | 0.670 |
| BLAB-SM | 0.904 | 0.770 | 0.647 | 0.677 | 0.800 | 0.570 | 0.825 | 0.881 | 0.753 | 0.783 | 0.838 | 0.695 |

**Table 11 Performance evaluation of all measures when NF = 3,000 – the averaged results (K = 1–120; +2).**

| Dataset | Reuters | | | | | | Web-KB | | | | | |
|---|---|---|---|---|---|---|---|---|---|---|---|---|
| Similarity/criterion | ACC | PRE | REC | FM | GM | AMP | ACC | PRE | REC | FM | GM | AMP |
| Euclidean | 0.652 | 0.576 | 0.228 | 0.242 | 0.451 | 0.216 | 0.431 | 0.627 | 0.293 | 0.218 | 0.473 | 0.281 |
| Cosine | 0.901 | 0.904 | 0.785 | 0.830 | 0.876 | 0.718 | 0.782 | 0.797 | 0.717 | 0.730 | 0.813 | 0.626 |
| Ex-Jaccard | 0.900 | 0.874 | 0.657 | 0.691 | 0.803 | 0.571 | 0.729 | 0.776 | 0.641 | 0.650 | 0.759 | 0.555 |
| Manhattan | 0.517 | 0.224 | 0.141 | 0.112 | 0.350 | 0.140 | 0.404 | 0.286 | 0.263 | 0.165 | 0.446 | 0.260 |
| KLD | 0.323 | 0.245 | 0.135 | 0.077 | 0.345 | 0.132 | 0.386 | 0.174 | 0.250 | 0.141 | 0.433 | 0.250 |
| SMTP | 0.907 | 0.895 | 0.703 | 0.748 | 0.830 | 0.632 | 0.807 | 0.854 | 0.724 | 0.741 | 0.819 | 0.653 |
| BLAB-SM | 0.902 | 0.889 | 0.670 | 0.725 | 0.809 | 0.603 | 0.816 | 0.876 | 0.726 | 0.742 | 0.821 | 0.665 |

highest ACC, FM, and AMP with cosine being superior over BLAB-SM and SMTP in terms of FM and AMP. Thus, the top performers were BLAB-SM, SMTP, and cosine.

In Table 10, on Reuters dataset, Cosine, followed by BLAB-SM and SMTP, obtained the highest ACC. Interestingly, Cosine had been superior to BLAB-SM and SMTP with FM and AMP. However, on Web-KB dataset, like Table 9, BLAB-SM, SMTP followed by

**Table 12 Performance evaluation of all measures when NF = 6,000 – the averaged results (K = 1–120; +2).**

| Dataset | Reuters | | | | | | Web-KB | | | | | |
|---|---|---|---|---|---|---|---|---|---|---|---|---|
| Similarity/criterion | ACC | PRE | REC | FM | GM | AMP | ACC | PRE | REC | FM | GM | AMP |
| Euclidean | 0.593 | 0.714 | 0.227 | 0.253 | 0.446 | 0.220 | 0.407 | 0.448 | 0.269 | 0.177 | 0.451 | 0.265 |
| Cosine | 0.891 | 0.900 | 0.786 | 0.829 | 0.876 | 0.716 | 0.785 | 0.795 | 0.728 | 0.739 | 0.819 | 0.632 |
| Ex-Jaccard | 0.904 | 0.882 | 0.667 | 0.700 | 0.809 | 0.584 | 0.739 | 0.787 | 0.652 | 0.658 | 0.767 | 0.565 |
| Manhattan | 0.521 | 0.327 | 0.151 | 0.129 | 0.362 | 0.150 | 0.398 | 0.263 | 0.260 | 0.159 | 0.443 | 0.257 |
| KLD | 0.308 | 0.254 | 0.131 | 0.069 | 0.339 | 0.130 | 0.387 | 0.176 | 0.251 | 0.140 | 0.434 | 0.250 |
| SMTP | 0.907 | 0.897 | 0.702 | 0.748 | 0.829 | 0.632 | 0.806 | 0.853 | 0.722 | 0.738 | 0.818 | 0.651 |
| BLAB-SM | 0.904 | 0.886 | 0.668 | 0.722 | 0.808 | 0.600 | 0.815 | 0.871 | 0.724 | 0.737 | 0.820 | 0.661 |

**Table 13 Performance evaluation of all measures when NF = the whole size (Reuters = 18,308, web-kb = 33,025 features) – the averaged results (K = 1–120; +2).**

| Dataset | Reuters | | | | | | Web-KB | | | | | |
|---|---|---|---|---|---|---|---|---|---|---|---|---|
| Similarity/criterion | ACC | PRE | REC | FM | GM | AMP | ACC | PRE | REC | FM | GM | AMP |
| Euclidean | 0.882 | 0.895 | 0.779 | 0.821 | 0.870 | 0.706 | 0.760 | 0.773 | 0.712 | 0.714 | 0.808 | 0.604 |
| Cosine | 0.884 | 0.898 | 0.779 | 0.823 | 0.871 | 0.709 | 0.760 | 0.773 | 0.712 | 0.714 | 0.808 | 0.604 |
| Ex-Jaccard | 0.904 | 0.883 | 0.666 | 0.700 | 0.809 | 0.583 | 0.760 | 0.800 | 0.679 | 0.685 | 0.787 | 0.591 |
| Manhattan | 0.527 | 0.378 | 0.167 | 0.151 | 0.379 | 0.164 | 0.398 | 0.223 | 0.258 | 0.155 | 0.441 | 0.256 |
| KLD | 0.301 | 0.255 | 0.129 | 0.065 | 0.337 | 0.128 | 0.387 | 0.177 | 0.251 | 0.140 | 0.434 | 0.250 |
| SMTP | 0.905 | 0.894 | 0.688 | 0.733 | 0.821 | 0.617 | 0.804 | 0.851 | 0.718 | 0.730 | 0.816 | 0.644 |
| BLAB-SM | 0.902 | 0.848 | 0.659 | 0.708 | 0.803 | 0.590 | 0.814 | 0.865 | 0.723 | 0.731 | 0.820 | 0.656 |

**Table 14 Performance evaluation of all measures when the average of averaged results is considered.**

| Dataset | Reuters | | | | | | Web-KB | | | | | |
|---|---|---|---|---|---|---|---|---|---|---|---|---|
| Similarity/ Criterion | ACC | PRE | REC | FM | GM | AMP | ACC | PRE | REC | FM | GM | AMP |
| Euclidean | 0.756 | 0.670 | 0.485 | 0.506 | 0.663 | 0.421 | 0.582 | 0.675 | 0.483 | 0.472 | 0.630 | 0.422 |
| Cosine | 0.854 | 0.729 | 0.636 | 0.662 | 0.778 | 0.558 | 0.745 | 0.759 | 0.684 | 0.698 | 0.788 | 0.588 |
| Ex-Jaccard | 0.804 | 0.654 | 0.509 | 0.529 | 0.688 | 0.435 | 0.581 | 0.629 | 0.494 | 0.493 | 0.641 | 0.429 |
| Manhattan | 0.691 | 0.511 | 0.356 | 0.358 | 0.554 | 0.307 | 0.490 | 0.544 | 0.370 | 0.324 | 0.536 | 0.339 |
| KLD | 0.482 | 0.442 | 0.186 | 0.164 | 0.402 | 0.168 | 0.397 | 0.348 | 0.295 | 0.204 | 0.472 | 0.280 |
| SMTP | 0.862 | 0.725 | 0.602 | 0.623 | 0.760 | 0.518 | 0.772 | 0.805 | 0.692 | 0.707 | 0.794 | 0.614 |
| BLAB-SM | 0.870 | 0.717 | 0.586 | 0.617 | 0.751 | 0.508 | 0.776 | 0.828 | 0.693 | 0.712 | 0.795 | 0.625 |

cosine obtained the highest ACC, FM, and AMP respectively. Thus, the top performers were BLAB-SM, SMTP, and cosine.

From Table 11, on Reuters dataset, SMTP, followed by BLAB-SM and cosine, obtained the highest ACC, FM, and AMP with cosine being superior over SMTP and BLAB-SM in terms of FM and AMP. However, on Web-KB dataset, BLAB-SM, followed by SMTP and

**Table 15 Purity (mostly known as "Accuracy") – K-means performance.**

| Similarity measure/Metric | K = 5 | | K = 10 | | K = Number of classes | |
|---|---|---|---|---|---|---|
| | Reuters – 18,308 features | Web-KB - 33,025 features | Reuters – 18,308 features | Web-KB - 33,025 features | Reuters – 18,308 features (K = 4) | Web-KB - 33,025 features (K = 8) |
| Euclidean | **0.6348979** | **0.6701596** | **0.7467169** | 0.6244344 | **0.6376284** | **0.6480114** |
| Cosine | 0.6200751 | **0.6546797** | 0.6641529 | 0.6263396 | 0.6727344 | 0.6225292 |
| Jaccard | 0.55571447 | **0.6468207** | 0.6736445 | **0.6663491** | 0.6222858 | 0.6320553 |
| Ex-Jaccard | 0.5942478 | 0.6434865 | **0.6970484** | 0.6261014 | 0.6358080 | **0.6618242** |
| KLD | 0.6091535 | 0.6170517 | 0.7253933 | 0.6477733 | **0.6618125** | **0.6525363** |
| Manhattan | 0.5648979 | 0.5901565 | 0.6368767 | 0.5845236 | 0.6076209 | 0.6281809 |
| BLAB-SM | **0.6313405** | 0.6037151 | **0.6965284** | **0.6954036** | **0.6459498** | **0.63824720** |
| SMTP | **0.6261864** | 0.6408668 | 0.6885970 | **0.6856275** | 0.6374983 | 0.63372231 |

cosine, obtained the highest ACC, FM and AMP respectively. Thus, the top performers were BLAB-SM, SMTP and cosine.

In Tables 12 and 13, on Reuters dataset, SMTP, followed by BLAB-SM and Ex-jaccard, obtained the highest ACC. Cosine, Euclidean, SMTP, and BLAB-SM were the best on FM and AMP. However, on Web-KB dataset, BLAB-SM, followed by SMTP and cosine, obtained the highest ACC, FM, and AMP with cosine being superior over BLAB-SM and SMTP in terms of FM only. Thus, the top performers were BLAB-SM, SMTP, Ex-Jaccard and cosine.

Finally, and most importantly in the averaged results case, in Table 14, BLAB-SM and SMTP followed by Cosine were the best measures on both datasets with BLAB-SM taking the lead over SMTP, and SMTP taking the lead over cosine. Surprisingly, cosine/Reuters was the top in terms of FM and AMP.

## Clustering results

We fixed K on 5, 10, and the number of actual classes of each dataset. The K-means was conditioned to stop after (50) iterations, or the stability has been reached after two successive cycles. Centroids were randomly picked in each iteration. The best measures based on Table 15, BLAB-SM went side by side with Euclidean followed by SMTP and KLD. The bolded values in Tables 15 and 16 signify the best values.

On the other hand, based on results of Table 16, Manhattan, BLAB-SM, SMTP followed By Cosine have been the best measures.

## DISCUSSION

In this section, we have investigated two key points: (1) the stability of each similarity measure over each dataset, (2) which features number each similarity measure has been the best.

## Measures stability

Recalling the results of Tables 6–14, 17 holds the most stable measures. In Table 17, R and W refer to both Reuters and Web-KB respectively.

**Table 16 Entropy – K-means performance.**

| Similarity measure/Metric | K = 5 | | K = 10 | | K = Number of classes | |
|---|---|---|---|---|---|---|
| | Reuters – 18,308 features | Web-KB - 33,025 features | Reuters – 18,308 features | Web-KB – 33,025 features | Reuters – 18,308 features (K = 4) | Web-KB – 33,025 features (K = 8) |
| Euclidean | **0.4181014** | **0.6083767** | 0.3324207 | 0.6338744 | 0.4066976 | 0.6489474 |
| Cosine | 0.5079926 | **0.6226576** | 0.3728674 | 0.6445081 | **0.3673066** | **0.6287172** |
| Jaccard | 0.4551138 | 0.6505793 | 0.4093773 | 0.6063160 | 0.4044194 | **0.6138461** |
| Ex-Jaccard | **0.4210397** | 0.6307335 | 0.3478137 | 0.6219528 | 0.4078592 | 0.6325090 |
| KLD | 0.5174439 | 0.6295514 | 0.3337954 | 0.6137706 | **0.3724656** | 0.6335062 |
| Manhattan | 0.4481032 | 0.6387609 | **0.3225643** | **0.6034765** | **0.3866542** | **0.6089534** |
| BLAB-SM | 0.4941582 | **0.6273114** | 0.3253195 | 0.5561895 | 0.4045278 | 0.6356724 |
| SMTP | **0.4447862** | 0.6486512 | **0.3147964** | **0.5373072** | 0.4038267 | 0.6341636 |

**Table 17 Measure stability status.**

| NF/Measure | Euclidean | | Cosine | | Ex-Jaccard | | Manhattan | | kullback Leibler | | SMTP | | BLAB-SM | |
|---|---|---|---|---|---|---|---|---|---|---|---|---|---|---|
| | R | W | R | W | R | W | R | W | R | W | R | W | R | W |
| 10 | | 3 | | 2 | | | 3 | 3 | 3 | 1 | | | | 3 |
| 50 | 2 | | 1 | 3 | | | 2 | | | | 1 | 3 | 1 | 3 |
| 100 | 2 | | | 3 | | | 2 | | | | 3 | 3 | 1 | 3 |
| 200 | | | 3 | 3 | | | | | | | 3 | 3 | 3 | 3 |
| 350 | | | 3 | 3 | | | | | | | 3 | 3 | 3 | 3 |
| 3000 | | | 3 | 3 | | | | | | | 3 | 3 | 3 | 3 |
| 6000 | | | 2 | 3 | 1 | | | | | | 3 | 3 | 3 | 3 |
| All features | 2 | 3 | 2 | 3 | 1 | | | | | | 3 | 3 | 1 | 3 |
| Sum | 6 | 6 | 14 | 23 | 2 | | 7 | 3 | 3 | 1 | 19 | 21 | 18 | 21 |
| Stable points | 12 | | 39 | | 4 | | 11 | | 4 | | 40 | | 39 | |

Table 17 shows that SMTP, BLAB-SM and cosine as the most stable measures with 40, 39, and 39 points respectively. It gives one plus (++1) for each measure when a measure has been superior in terms of any criterion (ACC, FM, and AMP). On Reuters, the competition was held between BLAB-SM, SMTP, cosine, and Ex-Jaccard. Moreover, it can be concluded, from Table 17, that BLAB-SM, SMTP, cosine can be used effectively for both low, middle and high dimensional datasets on each NF. Euclidean and Manhattan also performed well on low dimensional datasets (NF in [10–200] features). EX-Jaccard was observed to behave well on the middle and high dimensional datasets (NF in [350–N] features).

## Performance analysis

In this section, using the accuracy, f-measure, and average mean precision, we analyze the measure's performance.

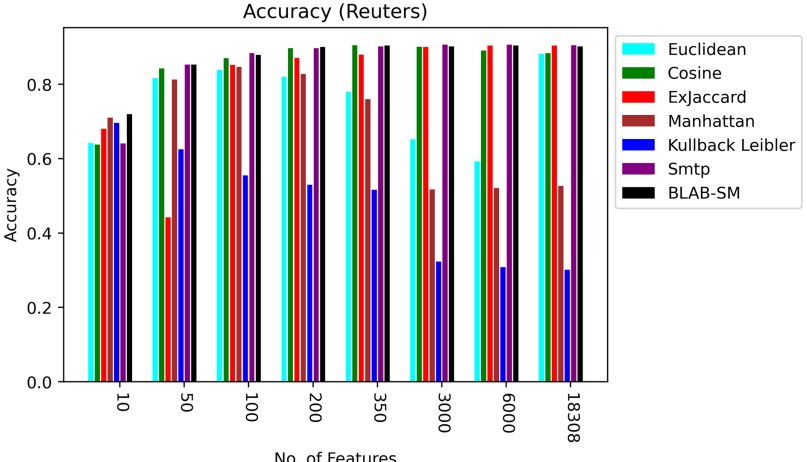

**Figure 5 Accuracy over all measures on all NF values – average results (K = 1–120; +2) – Reuters.**

## Reuters

Figure 5 shows BLAB-SM, SMTP, Ex-Jaccard, and Cosine achieved the most stable performance. The Cosine and ex-Jaccard showed a punctuated accuracy as cosine was superior when NF was in the range [10–3,000]. Ex-Jaccard had the lead as NF grew, though. When NF = 6,000, competition restricted among Cosine, Ex-jaccard, SMTP, and BLAB-SM with SMTP, BLAB-SM, and Ex-Jaccard being the best. However, as NF exceeded 6,000 features, the competition was held between SMTP, BLAB-SM, and Cosine with SMTP and BLAB-SM being the top performers. On the other hand, Manhattan, Euclidean, and kullback Leibler had the worst performance chiefly as NF grew. Albeit that their performance is good when the number of features (NF) were between 10 and 350, performance deteriorated steadily as NF surpasses 350.

In general, based on Figures 6 and 7, it can be concluded that Cosine, BLAB-SM, and SMTP had an almost close performance with slight superiority given to Cosine. Euclidean was shown a competitor with cosine when all features were considered, though. As expected, KLD, Manhattan, Euclidean had poor performance compared to BLAB-SM, SMTP, and Cosine. The Ex-jaccard had a competitive performance on Reuters, though.

## Web-Kb

Figures 8–10 illustrate the map of the criterion movements (results were averaged) for all measures over several NF values. On one hand, in Fig. 8, Cosine had been the middle ground between the first group of measures (KLD, Manhattan, Euclidean, and Ex-Jaccard) and those with the best performance (BLAB-SM and SMTP). BLAB-SM and SMTP had been involved in fierce competition with SMTP being superior as NF was in the range [10–100]. However, BLAB-SM had shown higher superiority as NF grew and surpassed 200 features. On the other hand, Fig. 8 reveals that Manhattan and kullback Leibler performed poorly as NF grew. Surprisingly, Ex-Jaccard had poor performance as NF was in the range [10–200]; yet, as NF grew, its performance grew and was competitive with Euclidean and cosine.

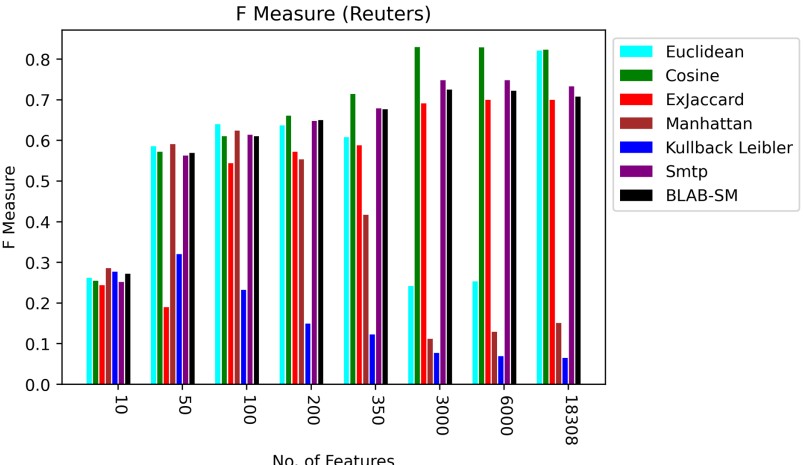

**Figure 6 F-measure over all measures on all NF values – average results (K = 1–120; +2) – Reuters.**

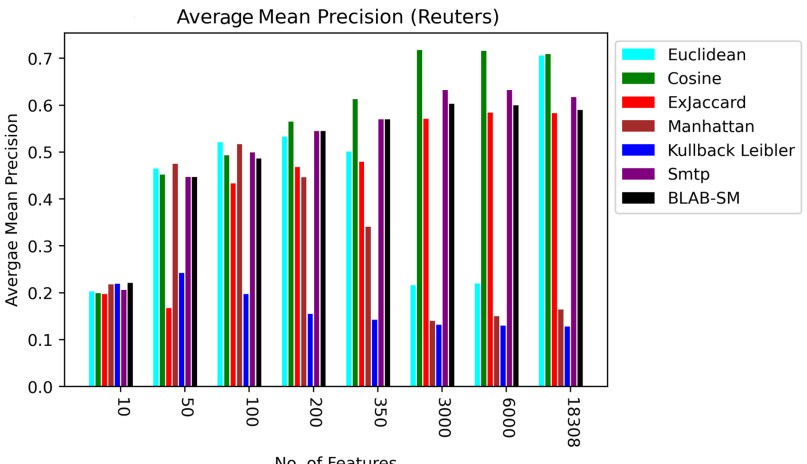

**Figure 7 AMP over all measures on all NF values – average results (K = 1–120; +2) – Reuters.**

Figures 9 and 10, on the other hand, show almost the same conclusion deduced from Fig. 7, with BLAB-SM being better than SMTP in terms of FM and AMP. In general, like Reuters, both BLAB-SM and SMTP had a close performance trend with significant superiority given to BLAB-SM regarding ACC and AMP. Interestingly, cosine was highly competitive with BLAB-SM and SMTP in terms of FM and AMP. Euclidean was an equivalent to cosine when all features were considered, though. Finally, KLD, Manhattan, Euclidean, and ex-Jaccard had lower performance on Web-KB compared with BLAB-SM, SMTP, and Cosine. Similarly to Reuters, Euclidean was superior to Ex-Jaccard when NF was in the range [10–350]. However, Ex-Jaccard had the lead when NF grew except for one case.

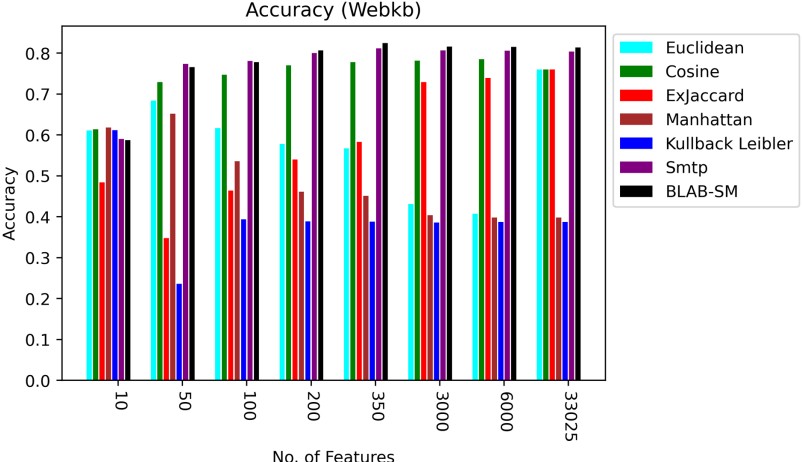

**Figure 8 Accuracy over all measures on all NF values – average results (K = 1–120; +2) – Web-KB.**

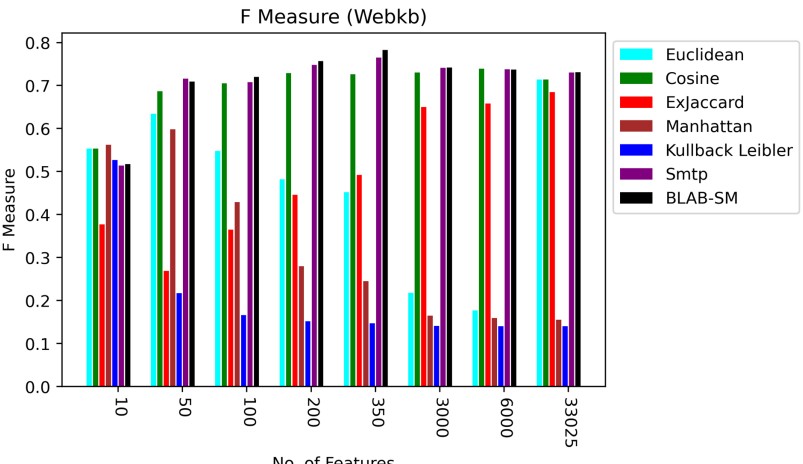

**Figure 9 F-measure over all measures on all NF values – average results (K = 1–120; +2) – Web-KB.**

Last but not least, Figs. 11 and 12 showcase the averaged performance of all measures on both datasets. Both figures stress that BLAB and SMTP have an almost equal performance trend as the top performers.

Finally, to provide the statistical evidence for the robustness of BLAB-SM's performance, the results of BLAB-SM against its rival measures have been verified on both datasets using the standard test of the non-parametric Wilcoxon test and the paired *t*-test. The statistical details for BLAB-SM, against all seven similarity measures, have been made based on the Accuracy "ACC" metric on both datasets, and the results are drawn in Tables 18–23. Results show that BLAB-SM is highly effective on Web-KB, and competitively effective with SMTP and Ex-Jaccard on Reuters. The statistical analysis is made by setting the standard value of the significance level at 0.05 (95%), and results are analyzed regarding the MAE, MSE, Std Error, z-score, *p*-value, and t-score. The degrees

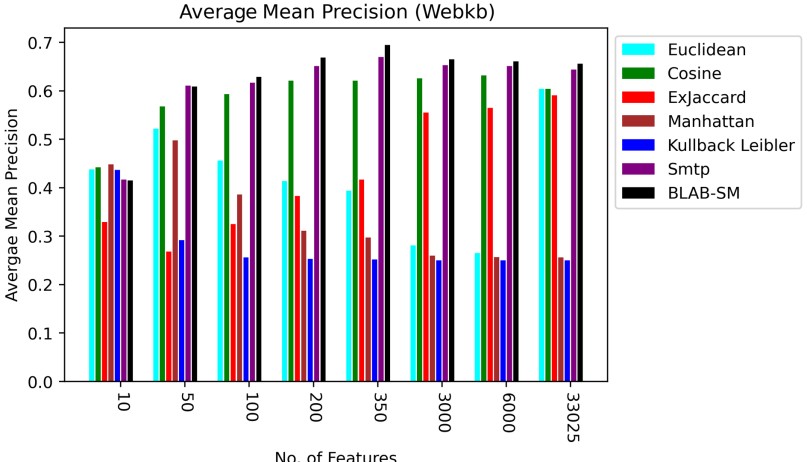

**Figure 10  AMP over all measures on all NF values – average results (K = 1–120; +2) – Web-KB.**

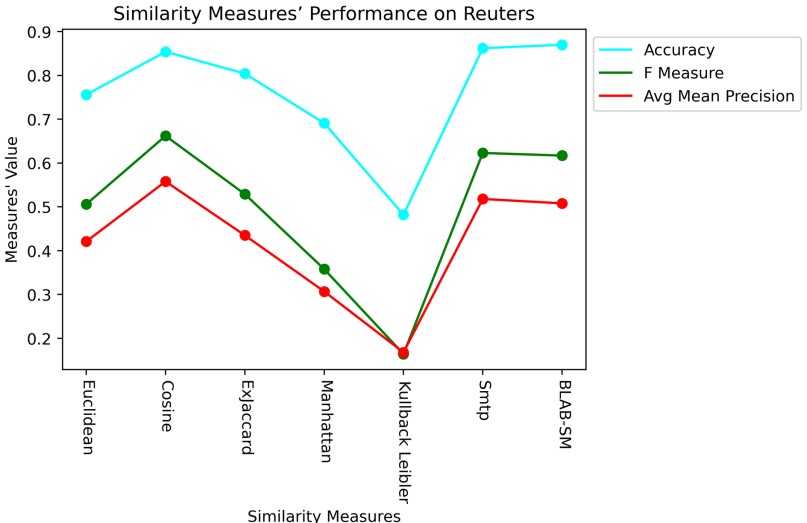

**Figure 11  Performance of all measures on both datasets – Reuters.**

of freedom (DF) for the paired *t*-test have been assigned the value of 59.0 as we have 60 K values (1–120, +2) for the KNN classifier (see Appendix). The negative sign of z-score and t-score implies the effectiveness and superiority of BLAB-SM comparing with its rival measures. In other words, both z-score and t-score indicate the significant difference between the BLAB-SM and its rivals.

Furthermore, for both datasets, the *p*-value of BLAB-SM is less than the value of the significance level (0.05). This is strong evidence for the performance robustness of BLAB-SM comparing with its rival measures. On one hand, based on the statistical results given in Tables 18–23, the negative sign of both z-score and t-score shows that the BLAB-SM measure is the top performer comparing with its rival measures. Moreover, z-score values of the BALB-SM measure are maximally significant in all cases as its

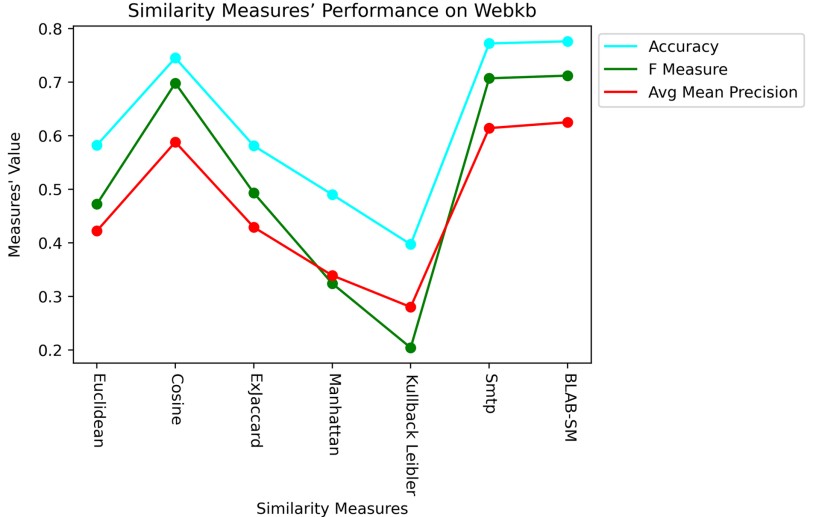

**Figure 12 Performance of all measures on both datasets – Web-KB.**

**Table 18 Reuters – statistical significance of the experimental results of BLAB-SM against all measures.**

| Measure | MAE | MSE | Std error | Confidence interval |
|---|---|---|---|---|
| Manhattan | 0.473236691 | 0.22496147 | 0.0040998006 | [0.51855962–0.53496698] |
| Euclidean | 0.117193134 | 0.01375278 | 0.0005560327 | [0.88169424–0.88391948] |
| Cosine | 0.114459829 | 0.01312222 | 0.0005939627 | [0.88435165–0.88672868] |
| Jaccard | 0.497886917 | 0.25102953 | 0.0072320509 | [0.48764178–0.51658438] |
| KLD | 0.496776287 | 0.24930169 | 0.0064743325 | [0.49026860–0.51617882] |
| SMTP | **0.095485359** | **0.00931860** | 0.0018312414 | **[0.90085033–0.90817894]** |
| Ex-Jaccard | **0.091511611** | **0.00845275** | 0.0011429639 | **[0.90620132–0.91077545]** |
| BLAB-SM | **0.099111785** | **0.01006081** | 0.0019902300 | **[0.89690577–0.90487065]** |

Note:
The smaller the values of MAE, MSE and Std error, the better the similarity measure is. Bolded numerical values in the BLAB-SM, SMTP and Ex-Jaccard rows are the best measure whose values of MAE, MSE and Std error are the smallest.

**Table 19 Reuters – statistical significance with non-parametric Wilcoxon for BLAB-SM against all measures.**

| Measure | Wilcoxon *Z* value | Wilcoxon *P* value |
|---|---|---|
| Manhattan | −6.7359808238 | 0.0 |
| Euclidean | −6.4488297578 | 0.00000000001 |
| Cosine | −5.6390451991 | 0.0000000171 |
| Jaccard | −6.7365741193 | 0.0 |
| KLD | −6.7365512974 | 0.0 |
| SMTP | −5.9849832403 | 0.0000000022 |
| Ex-Jaccard | −4.9985622992 | 0.0000005776 |
| BLAB-SM | Nan | nan |

**Table 20 Reuters – statistical significance with paired *t*-test (DF is degrees of freedom) for BLAB-SM against all measures.**

| Measure | t-score | tTest *p*-value | DF |
|---|---|---|---|
| Manhattan | −124.5610100138 | 0.0 | 59.0 |
| Euclidean | −8.2792339235 | 0.0 | 59.0 |
| Cosine | −6.8900087347 | 0.0000000042 | 59.0 |
| Jaccard | −49.7455369193 | 0.0 | 59.0 |
| KLD | −54.7808677782 | 0.0 | 59.0 |
| SMTP | 10.3197557538 | 0.0 | 59.0 |
| Ex-Jaccard | 5.6215858102 | 0.000000543 | 59.0 |
| BLAB-SM | Nan | nan | 59.0 |

**Table 21 Web-KB – significance test of BLAB-SM against all measures.**

| Measure | MAE | MSE | Std error | Confidence interval |
|---|---|---|---|---|
| Manhattan | 0.602047015 | **0.362962271** | 0.002891546 | [0.39216701–0.40373895] |
| Euclidean | 0.239183835 | 0.058129364 | 0.003916753 | [0.75297875–0.76865356] |
| Cosine | 0.238087691 | 0.057424802 | 0.003509637 | [0.75488953–0.76893507] |
| Jaccard | 0.776096143 | 0.602833150 | 0.002909542 | [0.21808187–0.22972583] |
| KLD | 0.777548864 | 0.605136971 | 0.003040655 | [0.21636679–0.22853547] |
| SMTP | **0.195668251** | **0.038413104** | **0.001455103** | **[0.80142009–0.80724340]** |
| Ex-Jaccard | 0.240649762 | 0.058737734 | 0.003709056 | [0.75192843–0.76677204] |
| BLAB-SM | **0.186283068** | **0.034801849** | **0.001294012** | **[0.81112761–0.81630624]** |

**Note:**
The smaller the values of MAE, MSE and Std error, the better the similarity measure is. Bolded numerical values in the BLAB-SM, SMTP and Ex-Jaccard rows are the best measure whose values of MAE, MSE and Std error are the smallest.

**Table 22 Web-KB – statistical significance with non-parametric Wilcoxon for BLAB-SM against all measures.**

| Measure | Wilcoxon *Z* value | Wilcoxon *P* value |
|---|---|---|
| Manhattan | −6.7365284757 | 0.0 |
| Euclidean | −6.7359351922 | 0.0 |
| Cosine | −6.7358895615 | 0.0 |
| Jaccard | −6.7368936511 | 0.0 |
| KLD | −6.7368936511 | 0.0 |
| SMTP | −6.213256068 | 0.0000000005 |
| Ex-Jaccard | −6.7358667465 | 0.0 |
| BLAB-SM | nan | nan |

*p*-values are less than (0.05) which is the level of significance. On the other hand, on Reuters only, in terms of t-score, all the results are significant except of SMTP and Ex-jaccard. Nevertheless, this slight insignificance of BLAB-SM could be compensated by the maximally-significant run time of BLAB-SM against both SMTP and Ex-Jaccard on Reuters.

**Table 23 Web-KB – statistical significance with paired _t_-test (DF stands for degrees of freedom).**

| Measure | t-score | tTest _p_-value | DF |
|---|---|---|---|
| Manhattan | −116.7083551117 | 0.0 | 59.0 |
| Euclidean | −13.9243714309 | 0.0 | 59.0 |
| Cosine | −15.2824025577 | 0.0 | 59.0 |
| Jaccard | −158.01549932 | 0.0 | 59.0 |
| KLD | −153.0210104997 | 0.0 | 59.0 |
| SMTP | −11.223396507 | 0.0 | 59.0 |
| Ex-Jaccard | −14.5841724659 | 0.0 | 59.0 |
| BLAB-SM | nan | nan | 59.0 |

**Table 24 Clustering process – points and rank of similarity measures.**

| Similarity measure | Points (out of 12) | Rank |
|---|---|---|
| Euclidean | 6 | 1 |
| Cosine | 5 | 2 |
| Jaccard | 3 | – |
| Ex-Jaccard | 3 | – |
| KLD | 5 | 2 |
| Manhattan | 4 | 3 |
| BLAB-SM | 6 | 1 |
| SMTP | 4 | 2 |

## Clustering analysis

Using the results of Tables 15 and 16, the analysis is made in Table 24. The points achieved by each similarity measure are accumulated on each corresponding metric. The whole number of points is 12 points. That is because of having two datasets and two metrics on three values of clustering variable (K = 5, K = 10, and K equal the number of actual classes).

## Execution time analysis

The time of each measure was accumulated and averaged as given in Figs. 13 and 14. It is clear that all measures share one fact, in particular Ex-Jaccard and SMTP, the run time is increasing sharply as NF increases steadily. Figures 13 and 14 give the run time of classification on Reuters and web-kb respectively.

As it is shown in Fig. 13, BLAB-SM followed by Euclidean and Jaccard were the fastest similarity measures with exception when NF=3000 in which KLD was faster. Manhattan followed by cosine and KLD came the next batch of the fastest measures. On the other hand, the slowest measures were SMTP and Ex-Jaccard with Ex-Jaccard being the slowest. Considering Fig. 13, BLAB-SM has been the sole measure that meets the ultimate quest of this work in terms of both effectiveness and efficiency at the same time. It is also worth indicating that the BLAB-SM has been as effective as SMTP and much better than Cosine. However, it is maximally efficient comparing with SMTP and more efficient than

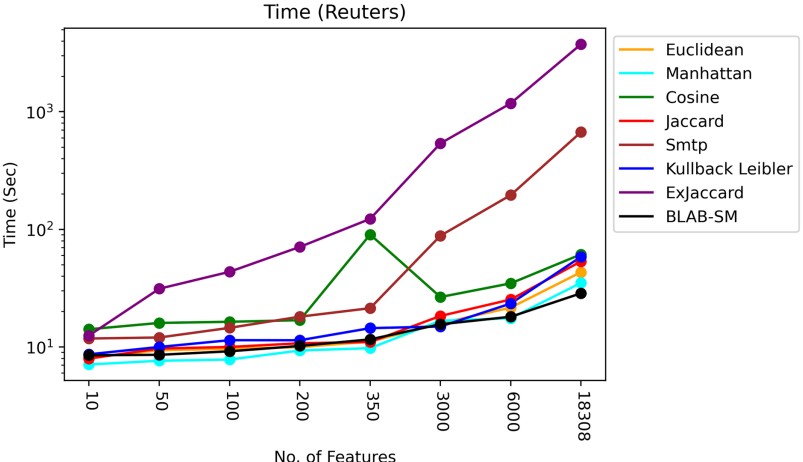

**Figure 13  Execution time – Reuters.**

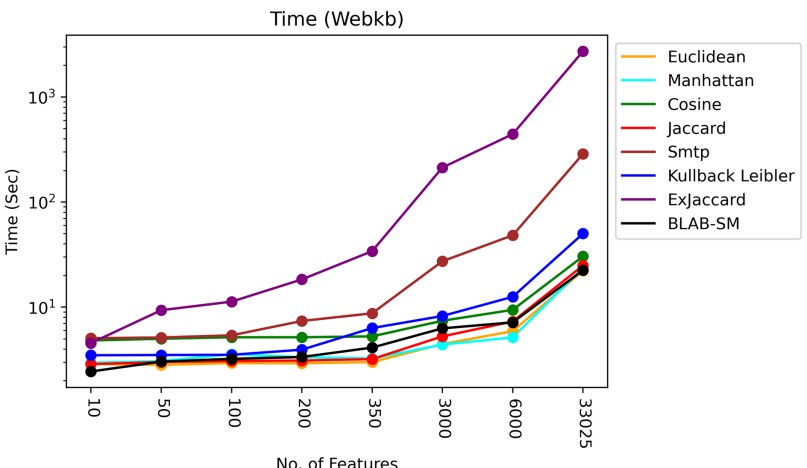

**Figure 14  Execution time – Web-KB.**

Cosine. Considering being faster, BLAB-SM could replace SMTP for text classification and clustering effectively and efficiently.

Also, Fig. 14 shows that Jaccard, Euclidean, BLAB-SM, and Manhattan were the fastest similarity measures. Interestingly, like Reuters, BLAB-SM confirms it is being fast measure on Web-KB as well. In general, Cosine was faster than KLD yet slower than BLAB-SM. On the other extreme, Ex-Jaccard and SMTP were seen to be slower measures with Ex-Jaccard being the slowest. In conclusion, according to all results drawn in Tables 8–16, and considering Fig. 14, BLAB-SM has been the sole measure that meets the ultimate quest of this work in terms of effectiveness and efficiency comparing with SMTP in particular and all measures in general. BLAB-SM has been as effective as SMTP, but maximally efficient compared to SMTP, and it also could replace SMTP and Cosine for text classification effectively and efficiently.

**Table 25 Comparison of classification run time (in minutes) of BLAB-SM against all measures on both datasets.**

| Measure/dataset | Reuters (18,308 features) | Improvement rate | Web-KB (33,025 features) | Improvement rate |
|---|---|---|---|---|
| BLAB-SM | 29 | – | 23 | – |
| Cosine | 58.28 | 50.24% | 26.37 | 12.78% |
| SMTP | 805 | 96.40% | 337 | 93.18% |
| Euclidean | 34.17 | 15.13% | 20.42 | −11.22% |
| Manhattan | 49.14 | 40.98% | 23.10 | 0.44% |
| Jaccard | 38.19 | 24.06% | 18.24 | -20.69% |
| kullback Leibler | 112.35 | 74.19% | 59.37 | 61.02% |

## Merits of BLAB-SM

In this subsection, it is worth referring briefly to the merits of our proposed measures over its competitors chiefly the SMTP. Concisely, the merits could be drawn as follows; (1) BLAB-SM has a simplistic design compared to SMTP. Moreover, BLAB-SM is bounded by upper and lower values in the same design with no complexity being added in the design, SMTP however needed an additional condition to restrict values of SMTP between zero and one (*Kumar Nagwani, 2015*). (2) BLAB-SM has been shown as effective as SMTP (and even better as it is the case on Web-KB) on the high dimensional dataset as all results asserted this claim when NF grew and exceeded 6,000 features on both datasets compared to SMTP. In other words, they both have an almost equal performance trend. Finally, (3) BLAB-SM has been observed to be impressively efficient compared to SMTP in particular and all state-of-art measures in general. Time Tables and graphs asserted this claim (see Table 25). When running measures on all features, the BLAB-SM has barely reached 29 min on Reuters and 23 min on web-KB compared to SMTP which needed roughly 804.933 min on Reuters and 336.567 min on Web-KB respectively. It is also noted that BLAB-SM is significantly faster than cosine which needed roughly 58.28 min on Reuters and 26.37 min on Web-KB respectively. With the BLAB-SM is being high-speed measure of a competitive performance, BLAB-SM could be nominated to replace Cosine and SMTP for the text-based applications, chiefly for the big "large-scale" datasets. The run time has been recorded when all features of both datasets were being considered. Table 25 summarizes the run time of all measures along with the BLAB-SM's improvement Rate against these measures.

## CONCLUSIONS AND FUTURE WORK

This work has developed a new similarity measure called Boolean Logic Algebra-Based Similarity measure (BLAB-SM) for the text-based applications. BLAB-SM has been designed to logically treat and process documents the same way the Boolean gates work. In the process of developing our BLAB-SM measure, a thorough experimental study has been conducted for several similarity measures for text classification and clustering. The results conclusively showed that the BLAB-SM similarity measure achieved a highly competitive performance on text retrieval quality (classification and clustering) and

run-time efficiency. The experimental results, revealed that BLAB-SM, SMTP, and Cosine scored the highest performance trends (with BLAB-SM and SMTP being the top performers) compared to Ex-Jaccard, Euclidean, Manhattan, and kullback Leibler measures. Ex-Jaccard showed a competitive performance on Reuters, though. On the other hand, Manhattan and kullback Leibler were seen to have the worst results. However, Ex-Jaccard and Euclidean had shown a fluctuating performance. They could be considered as middle-ground solutions between the best and worst measures with Ex-Jaccard being superior to Euclidean.

Jaccard has been experimentally proven ineffective choice when dealing with TF-IDF-based document matching. Given this fact, Jaccard was excluded from any further comparison with the state-of-the-art measures. That is because Jaccard depends basically on the rate of common features which is far less in the TF-IDF VSM matrix. Nevertheless, Jaccard has long been an effective measure when dealing with BoW based document matching. However, Jaccard showed a good clustering performance. BLAB-SM, Euclidean, SMTP, and Cosine were noted to be the top performers in terms of clustering, with BLAB-SM and Euclidean being the best.

In conclusion, run time comparison showed that BLAB-SM is one of the fastest similarity measures compared to all benchmarked measures in this work, making it highly promising in the machine learning and text mining fields. SMTP and Ex-Jaccard measures had been able to achieve good results; yet, there are efficiency costs as they were slow, Ex-Jaccard being the slowest measure. BLAB-SM is significantly efficient compared to SMTP while both have constantly been the most effective similarity measures.

The follow-up work is planned to consider the semantic aspect along with other similarity measures (*Amer & Abdalla, 2020*; *Oghbaie & Mohammadi Zanjireh, 2018*; *Sohangir & Wang, 2017*; *Aryal et al., 2019*) in an experimental study. Extensive experiments are planned to be conducted on a large-scale datasets to verify the conclusions of this work.

## ACKNOWLEDGEMENTS

The authors would like to thank and appreciate the support received from the Research Office of Zayed University for providing the necessary facilities to accomplish this work.

### Funding

This research has been supported by Research Incentive Fund (RIF) Grant Activity Code: R19093 – Zayed University, UAE. The funders had no role in study design, data collection and analysis, decision to publish, or preparation of the manuscript.

### Grant Disclosures

The following grant information was disclosed by the authors:
Zayed University, UAE: R19093.

## Competing Interests

The authors declare that they have no competing interests.

## Author Contributions

- Hassan I. Abdalla conceived and designed the experiments, analyzed the data, authored or reviewed drafts of the paper, and approved the final draft.
- Ali A. Amer conceived and designed the experiments, performed the experiments, analyzed the data, performed the computation work, prepared figures and/or tables, authored or reviewed drafts of the paper, and approved the final draft.

## Data Availability

The Python codes are available at GitHub: https://github.com/aliamer/Boolean-Logic-Algebra-Driven-Similarity-Measure-for-Text-Based-Applications.

The datasets are available at GitHub:

https://github.com/aliamer/Boolean-Logic-Algebra-Driven-Similarity-Measure-for-Text-Based-Applications/blob/main/Reuters%20%2B%20WebKB%20datasets.rar.

## Supplemental Information

Supplemental information for this article can be found online at http://dx.doi.org/10.7717/peerj-cs.641#supplemental-information.

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
