# Peer review of "Boolean logic algebra driven similarity measure for text based applications"

_PeerJ Computer Science, doi:10.7717/peerj-cs.641_

## Round 0.1 · original submission · Major Revisions

The paper should be carefully revised based on the comments of the reviewers.

Reviewer 1 ·

Basic reporting

1. There are so many errors and ambiguous expressions. The paper needs to further improve the language.
2. The structure should be further improved.
3. The motivation and contribution are not clear and enough. The necessity and effectiveness of your proposed approach should be highlighted. The literature references should be added.
4. The related work is not comprehensive. This part should review the latest research progress and the related information. The details of the different measurement methods are not related work, and they should be in the basic methods.

Experimental design

1. The experimental design flow chart can be supplemented to improve the readability of this paper.
2. The experimental results should be added quantitative comparative analysis, not only which method is better.
20. The detailed discussion should be further added.

Validity of the findings

1. The novel similarity measurement method is one of the hot topics in the current text mining. The research is meaningful, but the motivation and contribution need to be further highlighted in this paper.
2. The experimental results' description needs to be supplemented to verify the validity of the findings.

Additional comments

A new similarity measurement method based on Boolean logic algebra (BLAB-SM) is proposed to calculate the distance between texts in this paper. Comprehensive experiments have been conducted in two datasets to show the new method's superiority in seven measures. However, there are still some problems that need to be improved.
1. The motivation and contribution are not clear and enough. The necessity and effectiveness of your proposed approach should be highlighted.
2. The related work is not comprehensive. This part should review the latest research progress and the related information. The details of the different measurement methods are not related work, and they should be in the basic methods.
3. Why do you use TF/IDF for feature selection instead of N-gram, word2vec, and other methods?
4. Why do you choose the KNN classifier and K-means clustering method? Please make a supplementary statement.
5. There are some expression and description errors in this paper, e.g., line 20, lines 22-25, lines 38, lines 163-177.
6. The abbreviation should follow the full name, followed by a detailed description of the method improvement section, e.g., line 101, STB-SM, line 105 SMTP, line 111 PDSM.
7. The details of the two vectors doc1 and doc2 should be added in line 121 to improve the readability of the following methods.
8. The variable descriptions of Pdoc1 and Pdoc2 should be added in line 155.
9. The formulas in 163-177 are poorly aligned.
10. The 'Where' in lines 179, 245, 422, 430 should be 'where'.
11. In line 180 'the variable var', where is this variable?
12. In line 240, the same expressions can be removed.
13. In line 244, the formula should be rewritten to be consistent, e.g., the bracket (), ti, tj and tj. Please modify the similar expressions in the full manuscript, e.g., lines 254-257.
14. There are so many formulas and variables description should be revised.
15. Please check lines 261-264, and lines 284-289, e.g., 2x in line 263, 2x in line 289. What's the meaning of x, is variable or multiplication?
16. In 323 f2 X f4, X should be the multiplication, not X.
17. The formula number should be added.
18. There are many TF/IDF, TF-IDF, and TFIDF, and they should be revised to the same expression.
19. The experimental results should be added quantitative comparative analysis, not only which method is better from line 470 to line 591.
20. The discussion should be further added in line 625.

The figures and tables are very important and should be shown directly in the manuscript as well. I think a complete manuscript is more conducive to review.

Reviewer 2 ·

Basic reporting

The article should include sufficient introduction and background to demonstrate how the work fits into the broader field of knowledge. Relevant prior literature should be appropriately referenced.

Experimental design

The submission should clearly define the research question, which must be relevant and meaningful. The knowledge gap being investigated should be identified, and statements should be made as to how the study contributes to filling that gap.

Validity of the findings

The findings of the paper is not validated effectively.

Additional comments

The problem being studied in the paper is basically important however the contribution of the paper is not enough.

·

Basic reporting

I read the paper with interest although I must admit that at times it was difficult for me to follow due to frequent grammar and spelling issues. Below, for purposes of illustration, I list the spelling and grammar issues I spotted in the abstract alone.
Replace “information retrieval (IR)” with “the information retrieval (IR)”
Replace “data mining (DM) and” with “data mining (DM), and” (i.e. introduce a serial comma).
Replace “the similarity measures” with “similarity measures”
Replace “core stone” with “cornerstone”
Replace “algorithms upon which their performance is completely dependent” with “algorithms. Indeed their performance is completely dependent on these measures.”
Replace “is still in the full swings” with “is still uncompleted”. Perhaps rather than state that the quest is still ongoing, it may be stronger to point to what is wrong with existing similarity measures (i.e. arguments pertaining to their inefficiency and lack of effectiveness).
“Fastest runt time”. Fastest relative to what?
Replace “and K-means” with “and the K-means”
The sentence starting with “Using TF-IDF” is long and difficult to follow. I would suggest breaking this up into two.
Replace “Obtained results illustrate” with “Our results indicate”
Delete “, as a promising measure,”
Replace “has told that” with “establishes that”
Replace “: (1) BLAB-SM, as the most efficient and significantly-effective similarity measure, could replace state-of-art measures for text-based applications” with “BLAB-SM is not only more efficient but also significantly more effective than state-of-the-art similarity measures.
Based on the above (and my reading of the manuscript itself), my first recommendation is to have your paper thoroughly proofread and edited for readability by a native speaker. As a further general style recommendation, I would recommend the authors please try to refrain from extreme statements “perfect techniques” (line 43), “indisputably” (line 647) or self-congratulatory remarks “results are drawn elegantly” (line 87), as in my experience such wording may lead the critical reader to want to find ways to disagree with that which is being claimed. Try to aim for a more neutral tone. Please also write out all acronyms in full the first time they are mentioned (e.g. SMTP on line 105).
Before initiating the review of similarity and distance measures in the related work section (line 91), it would be useful to the reader to introduce a brief section detailing the various criteria on which the quality of similarity measures can be gauged and compared, before systematically reviewing the strengths and weaknesses of each along these criteria. All the claims about effectiveness and efficiency in this section are currently difficult to interpret because either it is unclear how effectiveness was operationalized, what other similarity measure(s) the focal measure is being compared to, or both (e.g., line 105: “shown highly effective when run against some similarity measures” is quite ambiguous to me).
Furthermore, I would suggest integrating the material in lines 93-122 in the paragraphs that follow, so that each of these paragraphs introduces the focal measures, elaborates how each is computed and discusses the evidence for the effectiveness and efficiency of each. It may also help to introduce a table in which each of these measures can easily be compared to one another.

Line 259 and line 266: Please replace “worst case” with something like “perfect dissimilarity”. Dissimilarity in my mind does not have to be a bad thing. Likewise, on p . 283 replace “the best case” with “Perfect similarity”

Experimental design

Other than stating on line 50 that little is known about the run time efficiency of existing measures, the introduction and related work section does not make a particularly compelling case for why we need yet another similarity measure. That being the case I felt the manuscript in its current form does not really succeed in establishing a knowledge gap.

Indeed, the fact that run time has not been studied cannot be used as input to the conclusion that such run times are bad. What are the gains to be expected from a similarity measure that is more effective and efficient? What exactly is the huge problem that the introduction of a new measure is going to resolve?

I feel that demonstrating how existing measures are suboptimal, along the lines I outline above (and beyond) would really strengthen the case for introducing BLAB-SM.

When it comes to the properties described as of line 296, please clarify at the outset that these properties complement one another and that one property may serve to decrease similarity while the other is simultaneously increasing it (for instance on the face of it property 1 and 2 oppose one another). To further clarify this, perhaps it may help to start the description of each property with “All else remaining the same, …”. Please also reference each property to the extant literature. Assuming I am understanding correctly, I think it would be clearer to frame property 3 in terms of the number of features that are present in both vectors. I did not understand property 6. Finally, I was missing a property pertaining to the scaling of the similarity measure. Clearly, a similarity measure must be bounded for it to be useful and interpretable across studies. Furthermore, I would expect a similarity measure to be independent of the number of features and documents (N), so that I can compare similarities obtained from different corpora.

Validity of the findings

Tables 1-3 (see p. 59-63) appear to be incomplete, in that I would expect to see all four possible combinations of A and B. Also please replace “table truth” with “truth table” in the title.

In the section on Machine Description, I found it curious the authors rely on software (Windows 7) that has been discontinued for some time now.
To generate the training and testing data (see line 281), I would recommend using k-fold cross-validation. Currently, it is unclear to me what the authors mean with “was divided individually and then combined as training and testing data.” Despite the data being well known, I would recommend including references to both datasets. Furthermore, and at minimum, I would like to see more detail on the datasets, in terms of size, origin, and descriptive data.
I would have liked to read more about why the authors decided to use KNN and K-means to validate BLAB-SM. Why these methods and not others? Furthermore, in the section entitled Experimental setup, I was missing a description of the KNN algorithm. Here at minimum, I would have liked to read a little more about what the classes were for each dataset, and how they were balanced.

In the section on the merits of BLAB-SM (line 898), I thought it may be worthwhile to reflect more on the merits defined in terms of reductions in energy consumption if BLAB-SM were to be adopted on a large scale.

Additional comments

Thank you for giving me the opportunity to review your paper, entitled “Boolean logic algebra-driven similarity measure for text-based applications”. The paper presents a novel similarity measure, that may augment or even replace existing measures because of its competitive efficiency and effectiveness metrics. I also appreciated the fact that the authors chose to focus on a problem as fundamental as defining a new similarity measure. I hope the authors will find my review useful in further strengthening their work.

---

## Round 0.2 · Minor Revisions

Please revise and check the whole paper carefully based on the comments of the reviewer.

Reviewer 1 ·

Basic reporting

1. There are also some minor errors in the paper, for example, Line 38 information retrieval (IR), data mining (DM), and machine learning (ML). They should be the same as the content in Abstract.
2. Line 238-250, 349-352, 375-379, 454-455, the format of these formulas are so bad. So please carefully revise the whole paper.
3. The whole structure still has a problem and not clear enough, especially the experimental results.

Experimental design

1. The structure of the experimental result is still needed to improve and make it more clear.

Validity of the findings

That's enough.

Additional comments

Please revise the whole paper and further improve the structure of the experimental results.

---

## Round 0.3 · accepted · Accept

I believe the paper was revised well.